# Spectral Graph Neural Networks are Incomplete on Graphs with a Simple Spectrum

**Snir Hordan**
Faculty of Mathematics
Technion - Israel Institute of Technology

**Maya Bechler-Speicher**
Meta

**Gur Lifshitz**
Blavatnik School of Computer Science
Tel-Aviv University

**Nadav Dym**
Faculty of Mathematics
Technion - Israel Institute of Technology

## Abstract

Spectral features are widely incorporated within Graph Neural Networks (GNNs) to improve their expressive power, or their ability to distinguish among non-isomorphic graphs. One popular example is the usage of graph Laplacian eigenvectors for positional encoding in MPNNs and Graph Transformers. The expressive power of such Spectrally-enhanced GNNs (SGNNs) is usually evaluated via the $k$-WL graph isomorphism test hierarchy and homomorphism counting. Yet, these frameworks align poorly with the graph spectra, yielding limited insight into SGNNs' expressive power. In this paper, we leverage a well-studied paradigm of classifying graphs by their largest eigenvalue multiplicity to introduce an expressivity hierarchy for SGNNs. We then prove that many SGNNs are incomplete even on graphs with distinct eigenvalues. To mitigate this deficiency, we adapt rotation equivariant neural networks to the graph spectra setting, yielding equiEPNN, a novel SGNN that provably improves upon contemporary SGNNs' expressivity on simple spectrum graphs. We then demonstrate that equiEPNN achieves perfect eigenvector canonicalization on ZINC, and performs favorably on image classification on MNIST-Superpixel and graph property regression on ZINC, compared to leading spectral methods.

## 1 Introduction

Graph Neural Networks (GNNs) have become a ubiquitous paradigm for learning on graph-structured data. The core principle of GNNs is to maintain a representation of each graph vertex and leverage the graph structure to iteratively refine each representation by its vertex's graph neighborhood [41]. To enhance the purview of the vertex's neighborhood, it is common to incorporate spectral features, such as Random Walk matrices, positional encoding, and graph distances, into the refinement operation of GNNs [9, 1, 45, 51]. Such GNNs, which systematically incorporate spectral features within their representation refinement procedure, or Spectrally-enhanced GNNs (SGNNs)[52], have gained significant traction in the graph learning community, due to their reasonable complexity and empirical benefits [18, 53, 52, 13].

Understanding the expressive power of GNNs provides researchers with a framework for comparing different models and identifying their deficiencies, often leading to improvements [15, 32, 35, 16, 49]. These frameworks ought to characterize which graphs the GNN can distinguish among, based on the GNNs' inner workings. For instance, the Weisfeiler-Leman (WL) test, which maintains and refines vertex representations similarly to Message Passing Neural Networks, a subclass of GNNs, completely determines which graphs these models can distinguish among [49].

39th Conference on Neural Information Processing Systems (NeurIPS 2025).

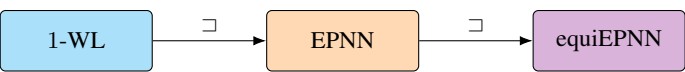

Figure 1: Hierarchy of 1-WL test variants. The arrows with ⊐ indicate strict inclusion relationships, meaning each variant can distinguish all graphs that the previous one can, plus additional graphs. Standard 1-WL is the least discriminative, while equiEPNN achieves the highest discriminative power, by incorporating both spectral invariant and equivariant refinement.

To study the expressive power of SGNNs, recent papers [52, 13] proposed a spectrally enhanced GNN, called Eigenspace Projection GNN (EPNN), which generalizes many popular spectral graph neural networks, and analyze its expressivity via WL tests and homomorphism counting. This comparison is valuable in comparing the expressivity of SGNNs to that of their combinatorial GNN counterparts. Yet, this analysis does not yield insight into the role of the graph spectra in the distinguishing ability of these GNNs.

To address this gap, we propose analyzing the expressive power of SGNNs via Spectral Graph Theory, and in particular via the maximal eigenvalue multiplicity of a graph. As isomorphism of graphs with bounded eigenvalue multiplicity can be determined in polynomial time, with the complexity depending exponentially on the eigenvalue multiplicity [2], this notion imposes a natural hierarchical classification of graphs, and SGNNs can potentially be complete on these graph classes, making this hierarchy a viable method for assessing their expressive power.

Our analysis centers around the expressivity of EPNN on graphs with distinct eigenvalues. This model is at least as expressive as many commonly used SGNNs [52], making an upper bound on the expressivity of EPNN applicable to these models. Surprisingly, we find that EPNN is incomplete even on the class of graphs with distinct eigenvalues. On the positive side, EPNN achieves completeness on simple spectrum graphs whose eigenvectors exhibit certain sparsity patterns. Based on these theoretical insights, we propose equiEPNN, inspired by equivariant neural networks for point clouds, which attains provably improved expressivity on graphs with distinct eigenvalues.

Our main contributions are summarized as follows:

1. We prove the incompleteness of EPNN (in Subsection 3.2) on graphs with a simple spectrum.

2. We formulate a guarantee on the completeness of EPNN on graphs with a simple spectrum based on sparsity patterns of the eigenvectors.

3. We introduce equiEPNN (in Section 3.3), a modified EPNN variant, which integrates Euclidean message passing into the feature refinement procedure.

4. We benchmark equiEPNN on the ZINC and MNIST-Superpixel datasets, yielding favorable performance in comparison with popular spectral methods. Furthermore, equiEPNN performs perfect eigenvector canonicalization on the ZINC dataset.

## 2 Related work

### 2.1 Spectral invariant GNNs

An enhancement to MPNNs and Transformer-based models is to incorporate spectral distances such as Random Walk, resistance, and shortest-path distances within the message passing operation [26, 50, 33, 12]. Zhang et al. [52] compare among spectral GNNs and the WL hierarchy by proving EPNN is strictly more powerful than 1-WL yet strictly less powerful than 3-WL. Despite the important result that 3-WL strictly bounds the expressive power of EPNN, the large expressivity gap between 1- and 3-WL makes this determination difficult to conceptualize. Building on this work, Gai et al. [13] have characterized the expressive power of EPNN via graph homomorphism counting, showing spectral invariant GNNs can homomorphism-count a class of specific tree-like graphs. Despite providing a deeper understanding of EPNN's expressive power, it remains hard to conceptualize and propose more expressive models based on it.

## 2.2 Spectral canonicalization methods

The eigenvectors of a graph are used as positional encoding to improve the expressive power of message-passing and as positional encoding for Transformer [38, 21, 31] based models. Yet, positional encoding has an inherent ambiguity problem. An eigenvector corresponding to a unique eigenvalue can be represented as itself or its negation [43]. Canonicalization methods [30, 29] are used to address the ambiguity problems of eigenvectors, by choosing a unique representative for each eigenvector.

Ma et al. [29] have uncovered an inherent limitation of canonicalization methods that process each eigenspace separately, which is that they cannot canonicalize eigenvectors with nontrivial self-symmetries. These models process each eigenbasis independently to obtain an orthogonal invariant and permutation-equivariant feature, and then use these features for downstream applications. Notable examples include SignNet and BasisNet [27], MAP [29] and OAP [30]. Ma et al. [30] have shown that these methods lose information when canonicalizing eigenvectors with self-symmetries, proving that the popular spectral invariant models SignNet and BasisNet are incomplete. In section 5.2, we provide a canonicalization scheme that bypasses this issue, and while not provable complete on all eigenvectors, empirically it canonicalizes all eigenvectors corresponding to distinct eigenvalues, in the ZINC [19] dataset.

## 2.3 Expressivity on simple spectrum graphs

An early study on the connection between GNNs and spectral features of the underlying graph studied the expressive power of CGNs [47]. They have proven that linear graph convolutional neural networks (GCNs) can map a graph signal to any chosen target vector, if the graph has distinct eigenvalues. Yet, this graph signal is sampled randomly and thus is not equivariant to permutations of the graph nodes, which may lead to degraded generalization, see Bechler-Speicher et al. [5].

For more related work see Appendix C.

# 3 Problem statement

## 3.1 Spectral graph decomposition

Graphs are typically represented by a matrix $A \in \mathbb{R}^{n \times n}$, where the $(i, j)$-th entry of the matrix encodes the relationship between node $i$ and $j$. This matrix could be the adjacency matrix, the normalized or un-normalized graph Laplacian, or a distance or Gram Matrix where the graph nodes have some underlying geometry.

A crucial principle in the design of graph neural networks is the notion of permutation invariance. Since graph nodes are not endowed with an intrinsic order, we would like to think of a matrix $A$ and its conjugation $PAP^T$ by a permutation matrix $P \in S_n$, as being equivalent. Graph neural networks respect this invariance constraint and produce a permutation-invariant function $f$ satisfying $f(A) = f(PAP^T)$. One popular method to design these functions exploits the eigendecomposition of the matrix $A$.

In the general case, we assume that $A$ has an eigenbasis $v^{(1)}, \ldots, v^{(n)}$ of vectors of norm one, which corresponds to real eigenvalues $\lambda_1, \ldots, \lambda_n$. This assumption holds when in the typical case where $A$ is a symmetric matrix (e.g., adjacency and Laplacian matrices), and also often holds in other settings (e.g., Random Walk matrix). This endows an alternative representation of the matrix $A$ with its own symmetries. Firstly, we note that each vector $Pv^{(q)}$ will be an eigenvector of $PAP^T$ with the same eigenvalue $\lambda_q$. Secondly, if $v^{(q)}$ is an eigenvector of norm one, then so is $-v^{(q)}$. When the eigenvalues of $f$ are pairwise distinct, then these are all the relevant ambiguities. This is referred to as the simple spectrum case. In the case of an eigenbasis of dimension $k$, the eigendecomposition ambiguity is defined by orthogonal transformations in $O_k$. In this paper, we will focus on the simple spectrum case. In this case, we define sign-invariant functions as follows

**Definition 1** (Sign Invariant functions). *For fixed natural $n$ and $K \leq n$, denote*

$$\mathcal{V}^K_{\text{simple}} = \{(V, \vec{\lambda}) \in \mathbb{R}^{n \times K} \oplus \mathbb{R}^K | \quad \lambda_1 > \lambda_2 > \ldots > \lambda_K\}.$$

*We say that $F : \mathcal{V}_{\text{simple}}^K \to \mathbb{R}^m$ is* sign invariant *if*

$$F(V, \vec{\lambda}) = F(PVS, \vec{\lambda}), \quad \forall P \in S_n, \quad S \in \{-1, 1\}^K$$

We note that in this definition, $V$ represents a $n \times K$ matrix whose $K$ columns represent the first $K$ eigenvectors $v^{(1)}, \ldots, v^{(K)}$ of $A$, and the notation $S \in \{-1, 1\}^K$ means that $S$ is a diagonal matrix whose diagonal is a vector in $\{-1, 1\}^K$.

The notion of sign invariant function was first introduced in [27], and was later discussed in [29, 30]. These papers discuss a collection of parametric functions $\mathcal{F} = \{f_\theta(V, \vec{\lambda}) \mid \theta \in \Theta\}$, such that for all parameters $\theta$ the function $f_\theta$ is sign invariant. To understand the expressiveness of these models, we formally define the notion of completeness on simple spectrum graphs.

**Definition 2** (Sign Invariant Separation). *For $K \leq n$, let $\mathcal{F}$ denote a collection of sign invariant functions defined on $\mathcal{V}_{\text{simple}}^K$, and let $\mathcal{D}$ be a subset of $\mathcal{V}_{\text{simple}}^K$. We say that $\mathcal{F}$ is **complete** on $\mathcal{D}$ if for any non-isomorphic pair $(V, \vec{\lambda})$ and $(U, \vec{\eta})$ in $\mathcal{D}$, there exists a function $f \in \mathcal{F}$ such that*

$$f(V, \vec{\lambda}) \neq f(U, \vec{\eta}).$$

Ideally, we would like $\mathcal{F}$ to be complete on all of the domain $\mathcal{V}_{\text{simple}}^K$. If $\mathcal{F}$ is complete, then by applying it to eigendecompostions of graphs with simple spectrum, we will obtain models which can separate all graphs with simple spectrum, up to permutation equivalence. The goal of this paper is to understand whether existing sign-invariant functions are complete.

## 3.2 EPNN

We will focus on a large family of sign invariant functions named *Eigenspace Projection GNNs (EPNN)*. This family of functions, introduced in Zhang et al. [52], was shown to generalize many spectral invariant methods such as Random Walk, resistance, and shortest-path distances [26, 50, 33, 12]. This method is based on a message passing like mechanism, where the spectral information is encoded by using the projection onto eigenspaces as edge features. In the simple spectrum case, this method can be formulated as follows:

For a given eigendecomposition $(V, \vec{\lambda}) \in \mathcal{V}_{\text{simple}}^K$, we we initialize a coloring for each 'node' $i \in [n]$ by

$$h_i^{(0)} = V_i \odot V_i, \tag{1}$$

where $V_i \triangleq V_{i,:}$ is the $K$ dimensional vector $[V_{i,:}(1), \ldots, V_{i,:}(K)]$ obtained by sampling all eigenvectors at the $i$-th node, and $\odot$ denotes elementwise multiplication. Importantly, this initialization is sign-invariant: while the global sign of each eigenvector is ambiguous, the product of two elements of the same eigenvector is not.

We next iteratively refine the node features via the update rule:

$$h_i^{(t+1)} = \text{UPDATE}_{(t)}\left(h_i^{(t)}, \vec{\lambda}, \{(h_j^{(t)}, V_i \odot V_j) \mid j = 1, \ldots, n\}\right) \tag{2}$$

Here and throughout $\{\cdot\}$ denote multisets (multiplicities are allowed) and the multiset notation implies that $\text{UPDATE}_{(t)}$ is required to be invariant to the order of the elements in the multiset.

Finally, we apply a global pooling operation to obtain a final permutation invariant representation

$$h_{\text{global}} = \text{READOUT}(\{h_i^{(T)} \mid i = 1, \ldots, n\}) \tag{3}$$

Once $\text{UPDATE}_{(t)}$ and READOUT functions are determined, this procedure determines a function $f(V, \vec{\lambda}) = h_{\text{global}}$ which is sign-invariant as in Definition 1. The collection of all such functions obtained by all possible choices of $\text{UPDATE}_{(t)}$ and READOUT functions is denoted by $\mathcal{F}_{\text{EPNN}}$.

## 3.3 Equivariant EPNN

In [13], the authors suggest methods based on higher order WL tests to boost the expressive power of spectral message passing neural networks. The complexity of these methods is considerably higher

than EPNN. In contrast, we will now suggest a method for increasing the expressive power of EPNN without significantly changing model complexity.

Our suggestions are based on constructions from neural networks for geometric point clouds. These neural networks operate on point clouds $X \in \mathbb{R}^{n \times d}$ (where in many applications $d = 3$) and each of the $n$ points in $\mathbb{R}^d$ represents a geometric coordinate. Models for such data are required to be invariant (or equivariant) to both permutations in $S_n$ and rotations in $O(d)$. This equivariant structure is similar to, but not identical to, the situation we have for graph eigecomposition: under the simple spectrum assumption, the symmetry transformations we are interested in is a single global permutation, and $K$ sign changes, which are rotations in $O(1)^K$. In the more general setting, we will have a single permutation and multiple rotations, whose dimension is determined by the multiplicity of each eigenvalue.

Via this analogy, we can look at spectral models for graphs from the perspective of point cloud networks. From this perspective, EPNN resembles geometric invariant networks, such as Schnet [42], which are based on simple invariant features. In contrast, [20] and [44] showed that, at least for point clouds, expressivity can be increased by recursively updating a rotation equivariant (in our scenario, sign equivariant) feature $v_i^{(t)}$ in parallel with the invariant feature $h_i^{(t)}$. Inspired by these observations, we suggest the following sign equivariant feature refinement procedure:

We use the same initialization $h_i^{(0)}$ as in Equation 1, and we initialize the equivariant feature $v_i^{(0)}$ to $v_i^{(0)} = V_i$. We then iteratively update these two features via

$$h_i^{(t+1)} = \text{UPDATE}_{(t,1)}\left(h_i^{(t)}, \vec{\lambda}, \left\{(h_j^{(t)}, v_i^{(t)} \odot v_j^{(t)}) \mid j = 1, \ldots, n\right\}\right)$$

$$v_i^{(t+1)} = v_i^{(t)} + \sum_{j=1}^{n} v_j^{(t)} \odot \text{UPDATE}_{(t,2)}(h_i^{(t)}, h_j^{(t)}, v_i^{(t)} \odot v_j^{(t)})$$

where $\text{UPDATE}_{(t,1)}$ is a multiset function, and $\text{UPDATE}_{(t,2)}$ maps its input to $\mathbb{R}^K$ so that the elementwise product in the equation above is well defined.

After running this procedure for $T$ iterations, we obtain an invariant global feature $h_{\text{global}}$ by aggregating the invariant node features $h_i^{(T)}$ using a READOUT function, as in (3). This gives us a sign invariant function $f(V, \vec{\lambda}) = h_{\text{global}}$. We name the class of all functions obtained by running this procedure with all different choices of update and readout functions equiEPNN.

We note that we can obtain EPNN models by setting $\text{UPDATE}_{(t,2)}$ to be the constant mapping to the zero vector. Accordingly, $\mathcal{F}_{\text{equi}}$ is at least as expressive as EPNN. In Section 4.4 we will show that it is strictly more expressive.

# 4 On the incompleteness of spectral graph neural networks

In this section, we analyze the expressive power of EPNN and equiEPNN on graphs with a simple spectrum. We first provide a counterexample to prove its incompleteness of EPNN on simple spectrum graphs. We then show that an equiEPNN can separate the counterexample, thus proving it is strictly more expressive than EPNN. Next we provide a subset of $\mathcal{V}_{\text{simple}}^K$ on which EPNN is complete. Finally, we discuss how our results imply the incompleteness of popular spectral GNNs even in the simple spectrum case.

## 4.1 EPNN is incomplete

We first introduce a pair of non-isomorphic eigendecompositions, $(V, \vec{\lambda})$ and $(U, \vec{\lambda})$ in $\mathcal{V}_{\text{simple}}^K$, which EPNN cannot distinguish, that is, it assigns them the same final feature after any number of refinement steps. In this construction $n = 12, K = 6$, and we fix the same choice of distinct eigenvalues $\vec{\lambda}$ for both examples. To define $V, U$, we denote

$$z_0 = \begin{pmatrix} 1 \\ 1 \end{pmatrix}, \quad z_1 = \begin{pmatrix} -1 \\ 1 \end{pmatrix}, \quad z_2 = \begin{pmatrix} 1 \\ -1 \end{pmatrix}, \quad z_3 = \begin{pmatrix} -1 \\ -1 \end{pmatrix}, \quad 0_2 = \begin{pmatrix} 0 \\ 0 \end{pmatrix},$$

and note that $z_0, \ldots, z_3$ are the four elements of the abelian group $\{-1,1\}^2$. Using these, we define $U, V$ via

$$U^T = \begin{pmatrix} z_0 & z_1 & z_2 & z_3 & 0_2 & 0_2 & 0_2 & 0_2 & z_0 & z_1 & z_2 & z_3 \\ z_0 & z_1 & z_2 & z_3 & z_0 & z_1 & z_2 & z_3 & 0_2 & 0_2 & 0_2 & 0_2 \\ 0_2 & 0_2 & 0_2 & 0_2 & z_0 & z_1 & z_3 & z_2 & z_0 & z_2 & z_1 & z_3 \end{pmatrix}$$

$$V^T = \begin{pmatrix} z_0 & z_1 & z_2 & z_3 & 0_2 & 0_2 & 0_2 & 0_2 & z_0 & z_1 & z_2 & z_3 \\ z_0 & z_1 & z_2 & z_3 & z_0 & z_1 & z_2 & z_3 & 0_2 & 0_2 & 0_2 & 0_2 \\ 0_2 & 0_2 & 0_2 & 0_2 & z_1 & z_0 & z_2 & z_3 & z_2 & z_0 & z_3 & z_1 \end{pmatrix}$$

We now show that $U, V$ are not isomorphic and cannot be separated by EPNN:

**Theorem 1.** *(Incompleteness of EPNN) The following statements hold:*

1. *$U$ and $V$ are not isomorphic under the group action of $S_{12} \times \{-1,1\}^6$.*

2. *EPNN cannot separate $U$ and $V$ after any number of iterations.*

3. *$U$ and $V$ have no non-trivial automorphisms.*

*Therefore, EPNN is incomplete on simple spectrum graphs.*

*Proof Idea.* To show $U, V$ are not isomorphic, we note that for any pair of permutation-sign matrices taking $U$ to $V$, the first four columns of $U^T$ must be mapped the first four columns of $V^T$. The same is true for columns $5 - 8$ and $9 - 12$. Considering the first four columns, we see that any sign matrix mapping them from $U$ to $V$ will be of the form $\mathrm{diag}(z, z, z')$ for $z, z' \in \{-1, +1\}^2$. The same argument for columns $5 - 8$ and $9 - 12$ gives sign patterns of the form $\mathrm{diag}(z', z, z_1 \cdot z)$ and $\mathrm{diag}(z, z', z_2 \cdot z)$, respectively. But there is no sign pattern satisfying these three constraints simultaneously.

We now explain the lack of separation of EPNN. We refer to the multiset of the multiplications of a column $i$ with all the other columns, as the column $i$'s purview. In the initial step, the purview of each column in the first 4-column block in $V^T$ and $U^T$, is identical, as the first $4$ columns exhibit a group structure with the multiplication operation. Thus, the hidden states of the first $4$ indices of $U^T$ and $V^T$ will be identical. By similar arguments, this holds for the remaining two blocks. Thus, after a refinement step, the nodes in each block cannot distinguish among those from other blocks, both in $U^T$ and $V^T$. Therefore, additional refinement procedures maintain identical representations for members of each index 'block' and corresponding blocks in $U^T$ and $V^T$. This implies EPNN cannot separate $U$ and $V$.

A full proof of the theorem is provided in the Appendix. $\qquad\square$

**Remark:** In many cases we are interested in eigenvalue decompositions of symmetric matrices, in which case the columns of $V, U$ (the rows of $V^T, U^T$) should be orthonormal. While our $V, U$ do not satisfy this condition, in the Appendix we show how they can be enlarged to yield a counterexample that has the same properties, and does have orthonormal columns.

### 4.2 When is EPNN complete?

The counterexample proves that there is an inherent limit to the expressive power of contemporary spectral invariant networks. We note that in this example $U, V$ had a significant number of zero entries. We now show that when $U, V$ each have at least one row without any zeros, EPNN will be complete (in particular, this condition always holds when the matrices $U, V$ have less than $n$ zero entries):

**Theorem 2** (EPNN Can Distinguish Dense Graphs with Distinct Eigenvalues)**.** *Let $\mathcal{D} \subseteq \mathcal{V}_{\mathrm{simple}}^K$ denote the set of $(V, \vec{\lambda})$ where $V$ has a row without zero entries. Then EPNN is complete on $\mathcal{D}$.*

*Proof.* By assumption, an index $i$ exists such that the $i$-th row of $V$ has no zeros. The hidden state $h_i^{(1)}$ after a signal iteration of EPNN (see (2)) can encode the eigenvalues $\vec{\lambda}$, the squared values of each coordinate of $V_i$, and the multiset of pairwise products $V_i \odot V_j$, as

$$h_i^{(1)} = (V_i \odot V_i, \{V_i \odot V_j \mid j = 1, \ldots, n\}) \tag{4}$$

To recover $V$ from $h_i^{(1)}$ up to symmetries, we can fix the sign ambiguity by choosing all coordinates of $V_i$ to be positive. We can then recover the remaining $V_j$ from the multiset in Equation 4. $\qquad \square$

This uncovers the inner workings of EPNN in processing simple spectrum graphs. Essentially, each entry can be normalized to represent a group element in $O(1)$, which acts as a local frame of reference, see [10] for more background, allowing us to reconstruct the eigenvectors up to sign symmetries.

### 4.3 Unique node identification via EPNN

A well-known mechanism for circumventing the limited expressive power of GNNs is by injecting unique node identifiers (IDs), which break the symmetries that hinder GNNs' separation ability [28, 14]. Popular approaches include random node initialization [5] and combinatorial methods [8], yet they are either limited by their discontinuity or break permutation equivariance. A natural question is whether the node features from EPNN are unique after finitely many iterations? If so, we have attained node IDs that do not break equivariance and change continuously with the eigendecomposition, alleviating the deficiencies of widely-used methods. We answer this question in the affirmative, provided the eigenvectors adhere to a sparsity pattern.

**Theorem 3.** *(EPNN for Unique Node Identifiers) Let $\mathcal{D} \subseteq \mathcal{V}_{\mathrm{simple}}^K$ denote the set of $(V, \vec{\lambda})$ where $V$ has no automorphisms, and has at most one zero per eigenvector. Then, one iteration of EPNN with injective UPDATE and READOUT functions assigns a unique identifier to each hidden node feature.*

*Proof.* By contradiction, assume that there exist distinct indices $i, j$ such that $h_i^{(1)} = h_j^{(1)}$. By the definition of EPNN, we have that

$$\{V_i \odot V_k\}_{k=1}^n = \{V_j \odot V_k\}_{k=1}^n \text{ and } V_i \odot V_i = V_j \odot V_j. \tag{5}$$

We deduce for the second equality that $|V_i^{(q)}| = |V_j^{(q)}|$ for all coordinates $q = 1, \ldots, K$. If for some $q$ we had $V_i^{(q)} = 0$, then also $V_j^{(q)} = 0$, in contradiction to the assumption that the $q$-th eigenvector has at most one zero entry. Thus all entries of $V_i$ and $V_j$ are non-zero.

Next, we deduce from Equation 5 and the fact that $|V_i^{(q)}| = |V_j^{(q)}| > 0$ for all $q$, that

$$\{(s_i^{(q)} V_k^{(q)})_{q=1}^K\}_{k=1}^n = \{(s_j^{(q)} V_k^{(q)})_{q=1}^K\}_{k=1}^n$$

where $s_i^{(q)} \in \{\pm 1\}$ and is defined as $\frac{V_i^{(q)}}{|V_i^{(q)}|}$ and $s_j^{(q)}$ is defined analogously. This means that there exists a permutation $\sigma$ which swaps $i$ with $j$, such that $s_i^{(q)} V_k^{(q)} = s_j^{(q)} V_{\sigma(k)}^{(q)}$ for all $k = 1, \ldots, n$ and $q = 1, \ldots, K$. Equivalently,

$$PVS_1 = VS_2 \implies PVS_1 S_2 = V \tag{6}$$

where $S_1$ and $S_2$ are diagonal matrices with $s_i^{(q)}$ and $s_j^{(q)}$, respectively, on the diagonals, and $P$ is the permutation matrix corresponding to $\sigma$. Since $P$ swaps $i$ with $j$, this is a non-trivial automorphism, in contradiction to the assumption. Thus $h_i^{(1)} \neq h_j^{(1)}$, as required. $\qquad \square$

### 4.4 equiEPNN is strictly more expressive than EPNN

We show that equiEPNN is strictly more powerful than EPNN, as it separates the pair $U$ and $V$ from Subsection 4.1, which EPNN cannot separate:

**Corollary 1.** *equiEPNN (see Section 3.3) can separate $U$ and $V$ after 2 iterations. Thus equiEPNN is strictly stronger than EPNN.*

*Proof Idea.* We show that after a single iteration, the equivariant update step can yield new matrices $U^{(t)}, V^{(t)}, t = 1$ which have no zeros. From Theorem 2, we know that a single iteration of EPNN, and hence also equiEPNN, is complete for such $U^{(t)}, V^{(t)}$, and thus two iterations of equiEPNN are sufficient for separation. $\qquad \square$

While equiEPNN is stronger than EPNN, the following result (proven in the appendix) shows that equiEPNN is also incomplete over simple spectrum graphs:

**Theorem 4.** *(Incompleteness of Equivariant EPNN) There exist $X, Y \in \mathbb{R}^{16 \times 6}$ such that the following statements hold:*

1. *$X$ and $Y$ are not isomorphic under the group action of $S_{16} \times \{-1, 1\}^6$.*

2. *Equivariant EPNN cannot separate $X$ and $Y$ after any number of iterations.*

*Therefore, Equivariant EPNN is incomplete on simple spectrum graphs.*

In the appendix we also explain how this counterexample can be extended so that $X, Y$ are orthogonal matrices which thus can form a full eigendecomposition of a real symmetric matrix.

## 4.5  Incompleteness of spectral GNNs

Theorem 1 proves that EPNN is incomplete on graphs with a simple spectrum. This spectral isomorphism test upper bounds the expressive power of many popular distance-based GNNs, which incorporate graph distances as edge features, such as Random Walk, PageRank, shortest path, or resistance distances [50, 26, 1, 45, 51]. Therefore, an immediate corollary of Theorem 1 follows:

**Corollary 2.** *Graphormer-GD [50], PRD-WL [26], DiffWire [1], and Random-Walk based GNNs[45, 51] are incomplete over graphs with a simple spectrum.*

In addition to this result, in the appendix we prove that the model proposed by Zhou et al. [53] is not universal on simple spectrum graphs.

**Proposition 3.** Vanilla OGE-Aug [53] is incomplete over graphs with a simple spectrum.

# 5  Experiments

Our goal in the experiments section is twofold: (a) statistically evaluate the validity of our bounded eigenmultiplicity approach for measuring expressivity and (b) empirically exemplify the utility of equiEPNN [1]. To meet the first goal, we statistically analyze the eigenvalue multiplicity in real-world datasets, and the number of non-zero entries in the eigenvectors, to compare these with our theoretical conditions for EPNN completeness. We find that while the sparsity conditions for EPNN completeness are satisfied on some real-world datasets (MNIST-Superpixel), they are not satisfied on datasets with more intricate symmetries (ZINC). For the second goal, we evaluate the utility of the equivariant features derived from equiEPNN on the task of eigenvector canonicalization [30]. Finally, we benchmark equiEPNN against leading spectral methods on the popular ZINC and MNIST-Superpixel datasets.

## 5.1  Dataset statistics

We surveyed several popular graph datasets and documented their graph spectral properties. The results are shown in Table 1. We find that the MNIST Superpixel [34] dataset is almost homogeneously composed of graphs with a simple spectrum, and we find that $(96.9\%)$ of the graphs in this dataset have a full row without zeros, implying that EPNN is complete on almost all graphs.

Other datasets, such as MUTAG, ENZYMES, PROTEINS and ZINC [19, 36], contain a substantial amount of graphs with eigenvalue multiplicity 2 and 3. Despite this, the number of eigenspaces of dimensions 2 and 3 is very low, averaging at around 1 per graph. On datasets with highly symmetric graphs, such as ENZYMES and PROTEINS, the graphs do not meet the sparsity condition of Theorem 2, thus EPNN will not necessarily faithfully learn the graph structure. This exemplifies the need for more expressive models that are complete on graphs with higher maximal eigenvalue multiplicity and sparse eigenvectors.

---

[1]Code is available at `https://github.com/IntelliFinder/equiEPNN`

Table 1: Graph Statistics Analysis Across Different Datasets (Eigenvalue Tolerance: $10^{-4}$)

| Dataset Name | MUTAG | ENZYMES | PROTEINS | MNIST | ZINC |
|---|---|---|---|---|---|
| **Dataset Overview** | | | | | |
| Number of Graphs | 188 | 600 | 1,113 | 60,000 | 10,000 |
| **Eigenvalue Characteristics** | | | | | |
| Graphs with Distinct Eigenvalues | 41.5% (78) | 34.8% (209) | 22.1% (246) | 99.9% (59,950) | 40.7% (4,072) |
| Graphs with Multiplicity 2 Eigenvalues | 58.5% (110) | 65.2% (391) | 77.9% (867) | – | 59.3% (5,928) |
| Graphs with Multiplicity 3 Eigenvalues | 19.1% (36) | 46.2% (277) | 57.9% (644) | – | 26.2% (2,617) |
| Avg. Number of Multiplicity 2 Eigenvalues | 0.74 | 1.01 | 1.24 | – | 1.282 |
| Avg. Number of Multiplicity 3 Eigenvalues | 0.26 | 0.58 | 0.71 | – | 1.105 |
| **Eigenvector Properties** | | | | | |
| Average Ratio of Zeros | 1.67 | 4.28 | 6.39 | 0.31 | 2.52 |
| Average Number of Zeros | 31.13 | 172.93 | 817.20 | 23.16 | 61.04 |
| Graphs with a Full Row | 75.0% (141) | 35.8% (215) | 37.1% (413) | 96.9% (58,077) | 64.5% (6,447) |
| Graphs with $\leq 1$ Zero per Eigenvector | 0.0% (0) | 6.3% (38) | 5.0% (56) | 20.2% (12,085) | 4.3% (430) |
| Graphs with Total Zeros $<$ Vertices | 29.8% (56) | 16.3% (98) | 14.3% (159) | 89.9% (53,873) | 13.0% (1,295) |
| Graphs Meeting Any Condition | 75.0% (141) | 35.8% (215) | 37.1% (413) | 96.9% (58,077) | 64.5% (6,447) |

## 5.2 Eigenvector canonicalization

Positional encoding is a cornerstone of graph learning using Transformer architectures, yet they suffer from the sign ambiguity problem [9]. It can be resolved by eigenvector canonicalization, which involves choosing a unique representation of each eigenvector. Yet, an inherent limitation of current canonicalization methods is that they are unable to canonicalize eigenvectors with nontrivial self-symmetries, often called uncanonicalizable eigenvectors [29, 30].

Table 2: Uncanonicalizable Graph Eigenvectors in ZINC (Subset) [19] as percentage of total eigenvectors of eigen-space dimension 1.

| Property | Percentage (%) |
|---|---|
| Sum to 0 | 11.15 % |
| Uncanonicalizable | 10.93 % |
| equiEPNN output sum to 0 | 0.0 % |
| Uncanonicalizable after equiEPNN | 0.0 % |

To overcome this limitation, we devise a method to choose a canonical representation of the original eigenvectors via the equivariant output of equiEPNN. The only requirement is that each vector in the equivariant output does not sum to $0$.

We test our hypothesis on a popular benchmark ZINC [19], and find that all the vectors in the equivariant output are canonicalizable and sum to zero, in contrast to the vectors from the eigendecomposition, where $10\%$ of them are uncanonicalizable. Furthermore, we devise a way to choose a canonical representation of the original eigenvectors via the equivariant output and describe this in the Appendix. The results are shown in Table 2.

Table 3: Results on ZINC and MNIST-Superpixel datasets. The values are the MSE for ZINC (Subset) and the accuracy for MNIST-Superpixel. Edge features are not used even if they are available in the datasets. For ZINC, all models use node labels. For MNIST-Superpixel, the model uses superpixel-intensive values and node degree as node features. Models have a budget of 30K free parameters for ZINC and 35K for MNIST.

| Category | Model | ZINC (MAE ↓) | MNIST-Superpixel (Acc.↑) |
|---|---|---|---|
| NN | MLP | $0.5869 \pm 0.025$ | $25.10\% \pm 0.12$ |
| MPNN | GCN | $0.3322 \pm 0.010$ | $52.80\% \pm 0.31$ |
| | GAT | $0.3977 \pm 0.007$ | $82.73\% \pm 0.21$ |
| | GIN | $0.3044 \pm 0.010$ | $75.23\% \pm 0.41$ |
| 3-WL | PPGN | $0.1589 \pm 0.007$ | $90.04\% \pm 0.54$ |
| Spectral | ChebNet | $0.3569 \pm 0.012$ | $\mathbf{92.08\% \pm 0.22}$ |
| | GNNML1 | $0.3140 \pm 0.015$ | $84.21\% \pm 1.75$ |
| | equiEPNN (Ours) | $\mathbf{0.2805 \pm 0.019}$ | $90.32 \% \pm 0.7$ |

### 5.3 Benchmarks: ZINC and MNIST

We evaluated equiEPNN on the image classification task MNIST-Superpixel [34], in which clustering of images is performed according to regions with similar pixel values, an algorithm creates a graph based on these regions, and each node is assigned a region-induced feature. We compared equiEPNN to leading spectral methods, all with a comparable parameter budget of $\approx 35K$ (see Table 3). We observe that it outperforms PPGN [32], which has cubic complexity, and GNNML1, which also processes the eigendecomposition of the graph. ChebNet outperforms all other methods, perhaps due to its handcrafted polynomial features.

We further evaluate equiEPNN via the standard regression task on the ZINC dataset of molecular graphs (we also tested eigenvector canonicalization on this same dataset). ZINC (Subset) has 12000 graphs with an average of 23.16 nodes per graph. We compare ourselves to leading methods with the standard $\approx 500K$ parameter budget and find that, out of the spectral methods, our method attains the best results, see Table 4.

Table 4: Results on ZINC.

| Method | Test Error (MAE ↓) |
|---|---|
| GIN | $0.526 \pm 0.051$ |
| GraphSage | $0.398 \pm 0.002$ |
| GCN | $0.384 \pm 0.007$ |
| GCN | $0.367 \pm 0.011$ |
| GatedGCN-PE | $0.214 \pm 0.006$ |
| MPNN (sum) | $0.145 \pm 0.007$ |
| PNA | $0.142 \pm 0.010$ |
| GT | $0.226 \pm 0.014$ |
| SAN | $0.139 \pm 0.006$ |
| Graphormer$_{\text{SLIM}}$ | $0.122 \pm 0.006$ |
| MPNN | $0.138 \pm 0.006$ |
| EPNN | $0.103 \pm 0.006$ |
| equiEPNN (Ours) | $\mathbf{0.099 \pm 0.001}$ |
| Subgraph GNN | $0.110 \pm 0.007$ |
| Local 2-GNN | $0.069 \pm 0.001$ |

## 6 Future Work

A key future goal is to devise spectral GNNs that achieve completeness on graphs with simple spectra, and higher eigenvalue multiplicities. One interesting direction is to use higher-order point cloud networks to process the eigenvectors [53]. We have shown that treating each eigenspace as a separate entity does not lead to universality (see Subsection 4.5). Thus, these high-order networks should process the eigenvectors as a single entity, but remain invariant only to the sign and basis symmetries.

**Acknowledgements** N.D. and S.H. were supported by ISF grant 272/23.

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

# Appendix

## A   Proofs

### A.1   Proof of Incompleteness of EPNN

**Theorem 1.** *(Incompleteness of EPNN) The following statements hold:*

    *1. $U$ and $V$ are not isomorphic under the group action of $S_{12} \times \{-1, 1\}^6$.*

    *2. EPNN cannot separate $U$ and $V$ after any number of iterations.*

    *3. $U$ and $V$ have no non-trivial automorphisms.*

*Therefore, EPNN is incomplete on simple spectrum graphs.*

*Proof.* For convenience, we recall the definitions of the point clouds $U$ and $V$:

We denoted the four elements of the abelian group $\{-1, 1\}^2$, and the zero vector in $\mathbb{R}^2$, by

$$z_0 = \begin{pmatrix} 1 \\ 1 \end{pmatrix}, \quad z_1 = \begin{pmatrix} -1 \\ 1 \end{pmatrix}, \quad z_2 = \begin{pmatrix} 1 \\ -1 \end{pmatrix}, \quad z_3 = \begin{pmatrix} -1 \\ -1 \end{pmatrix}, \quad 0_2 = \begin{pmatrix} 0 \\ 0 \end{pmatrix}.$$

Using these, we define $U, V$ via

$$U^T = \begin{pmatrix} z_0 & z_1 & z_2 & z_3 & 0_2 & 0_2 & 0_2 & 0_2 & z_0 & z_1 & z_2 & z_3 \\ z_0 & z_1 & z_2 & z_3 & z_0 & z_1 & z_2 & z_3 & 0_2 & 0_2 & 0_2 & 0_2 \\ 0_2 & 0_2 & 0_2 & 0_2 & z_0 & z_1 & z_3 & z_2 & z_0 & z_2 & z_1 & z_3 \end{pmatrix}$$

$$V^T = \begin{pmatrix} z_0 & z_1 & z_2 & z_3 & 0_2 & 0_2 & 0_2 & 0_2 & z_0 & z_1 & z_2 & z_3 \\ z_0 & z_1 & z_2 & z_3 & z_0 & z_1 & z_2 & z_3 & 0_2 & 0_2 & 0_2 & 0_2 \\ 0_2 & 0_2 & 0_2 & 0_2 & z_1 & z_0 & z_2 & z_3 & z_2 & z_0 & z_3 & z_1 \end{pmatrix}$$

We first prove: **2**. the inseparability of $U$ and $V$ by EPNN.

Observe the purview of node $i$ of $U$ after the first refinement step of EPNN:

$$h_i^{(1)}(U) = (U_i \odot U_i, \{U_i \odot U_j \mid j \in [10]\}) \tag{7}$$

We will show that point clouds can be partitioned into 'blocks' such that each point in the block obtains the same hidden state. This block structure is recognized by viewing each point as a group element and each block as a multiplicative group. We will then show that this multiplicative group structure allows us to prove the inseparability of EPNN.

Concretely, our proof proceeds as follows:

1. The column entries of $U^T$ and $V^T$ can be partitioned into 3 blocks : $B_1 \triangleq \{1, 2, 3, 4\}, B_2 \triangleq \{5, 6, 7, 8\}$, and $B_3 \triangleq \{9, 10, 11, 12\}$, such $h_i^{(1)}(U)$ and $h_j^{(1)}(U)$ are identical for every $i, j \in B_k, k = 1, 2, 3$.

2. It holds that $h_i^{(1)}(U) = h_i^{(1)}(V)$ for every $i = 1, 2, \ldots, 12$.

3. For any $t \in \mathbb{N}$, $h_i^{(t)}(U) = h_i^{(t)}(V)$ for every $i = 1, 2, \ldots, 12$.

1. We first focus on $B_1$ and then extend the argument to $B_2$ and $B_3$.

Since the elements $U_j$ for $j = 1, 2, 3, 4$ admit a multiplicative group structure, then for every $i = 1, 2, 3, 4$, the respective entries $U_i \odot U_j$ for $j \in [4]$ are identical (closure of groups.)

For $j = 5, 6, 7, 8$ and $i = 1, 2, 3, 4$, the entries of the products $V_i \odot V_j$, are are zeros in two row entries and the non-zero entries in the remaining row, each element of the group $\mathbb{Z}_2^2 \cong \{z_0, z_1, z_2, z_3\}$ appears exactly once in the non-zero entries of the products, as it holds that $z_i \mathbb{Z}_2^2 = \mathbb{Z}_2^2$.

Analogously, we can extend this argument to $j = 9, 10, 11, 12$ and $i = 1, 2, 3, 4$.

This means that $h_i^{(1)}(U)$ and $h_j^{(1)}(U)$ are identical for every $i, j \in B_1$.

Since, by definition of $U$ and symmetry, each four-index quadruple $B_1 \triangleq 1, 2, 3, 4$, $B_2 \triangleq 5, 6, 7, 8$, and $B_3 \triangleq 9, 10, 11, 12$ is a multiplicative group, the analysis for the hidden states of the indices in $B_1$ holds for $B_2$ and $B_3$. This concludes item 1.

2. Up to now, we proved for indices $i, j \in B_k$ for $k = 1, 2, 3$, it holds that $h_i^{(1)} = h_j^{(1)}$. It remains to be proven that these hidden states are equivalent in both point clouds to conclude step 2.

Since the point cloud $V^T$ is derived from $U^T$ by multiplying the columns in $B_2$ by $\mathrm{diag}(z_0, z_0, z_1)$ and the columns in $B_3$ by $\mathrm{diag}(z_0, z_0, z_2)$, the purview (see Equation 7) of each index is identical in both point clouds, since $z_2 \cdot z_2 = z_1 \cdot z_1 = z_0$ which is the identity element, thus by definition of EPNN, this modification that maps $U^T$ to $V^T$ doesn't affect the pairwise multiplications in Equation 7.

3. To prove this step, we only need to show that the hidden states remain identical within each block, since the fact that they are identical across the point clouds stems from the same justification of step 2.

In the second update step, the arguments of Step 1 remain identical. Still, now we have updated hidden node information, but the hidden node information is identical across nodes belonging to the same block. Therefore, the only information this refinement yields is the categorization of nodes into blocks. Yet this information is already known in the initialized hidden states, $\{h_i^{(0)} \ i = 1, \ldots, n\}$, since the zero entries of multiplication $h_i^{(0)} = V_i \odot V_i$ determine the block that $i$ belongs to. Therefore, the hidden states don't supply the network with any supplementary information other than the initialization $h_i^{(0)} = V_i \odot V_i$. Thus, after a second refinement step, the hidden states remain identical within each block, as they have after the first refinement step. Moreover, the corresponding hidden states of the two point clouds also remain equivalent due to the arguments in Step 2, which remain analogous, as the hidden states after a refinement only assign each node its respective block

membership, which is exactly the information given in the first update step. This argument can then be applied recursively to any number of update steps.

In conclusion, we have shown that for any $t \in \mathbb{N}$, the hidden states of both point clouds are identical (in corresponding indices), therefore after a permutation invariant readout, we obtain the same output.

We now prove **3**. *U and V have no nontrivial automorphisms.* To show $U, V$ are not isomorphic, we note that for any pair of permutation-sign matrices taking $U$ to $V$, the first four columns of $U^T$ must be mapped the first four columns of $V^T$. The same is true for columns $5-8$ and $9-12$. Considering the first four columns, we see that any sign matrix mapping them from $U$ to $V$ will be of the form $\mathrm{diag}(z, z, z')$ for $z, z' \in \{-1, +1\}^2$. The same argument for columns $5-8$ and $9-12$ gives sign patterns of the form $\mathrm{diag}(z', z, z_1 \cdot z)$ and $\mathrm{diag}(z, z', z_2 \cdot z)$, respectively. But there is no sign pattern satisfying these three constraints simultaneously.

The automorphism group of this extended eigendecomposition is contained within that of $U$ and $U$, respectively, and thus is also only the trivial group.

The proof of **1**. which states that $U$ and $V$ are not isomorphic, is analogous to the proof of **3**, and yields that the only sign pattern taking each point cloud to itself is $(z_0, z_0, z_0)$, which implies each point cloud has only a trivial automorphism..

## A.2 Extension to orthonormal counterexamples

The rows of the above point clouds $U, V$ are not orthonormal. Thus, they are not eigenvectors of an eigendecomposition of a symmetric matrix. We fix this misalignment via the following 'orthogonalization' matrices:

Taking $\tilde{U}$ to be a concatenation of the previous $U$ and $\hat{U}$ defined by

$$\hat{U}^T = \begin{pmatrix} 2z_0 & 2z_1 & 2z_2 & 2z_3 & 0_2 & 0_2 & 0_2 & 0_2 & -2z_0 & -2z_1 & -2z_2 & -2z_3 \\ -\frac{z_0}{2} & -\frac{z_1}{2} & -\frac{z_2}{2} & -\frac{z_3}{2} & 2z_0 & 2z_1 & 2z_2 & 2z_3 & 0_2 & 0_2 & 0_2 & 0_2 \\ 0_2 & 0_2 & 0_2 & 0_2 & -\frac{z_0}{2} & -\frac{z_1}{2} & -\frac{z_3}{2} & -\frac{z_2}{2} & \frac{z_0}{2} & \frac{z_2}{2} & \frac{z_1}{2} & \frac{z_3}{2} \end{pmatrix}$$

Then take $\tilde{V}$ to be a concatenation of the previous $V$ and $\hat{V}$ defined by

$$\hat{V}^T = \begin{pmatrix} 2z_0 & 2z_1 & 2z_2 & 2z_3 & 0_2 & 0_2 & 0_2 & 0_2 & -2z_0 & -2z_1 & -2z_2 & -2z_3 \\ -\frac{z_0}{2} & -\frac{z_1}{2} & -\frac{z_2}{2} & -\frac{z_3}{2} & 2z_0 & 2z_1 & 2z_2 & 2z_3 & 0_2 & 0_2 & 0_2 & 0_2 \\ 0_2 & 0_2 & 0_2 & 0_2 & -\frac{z_1}{2} & -\frac{z_0}{2} & -\frac{z_2}{2} & -\frac{z_3}{2} & \frac{z_2}{2} & \frac{z_0}{2} & \frac{z_3}{2} & \frac{z_1}{2} \end{pmatrix}$$

The columns of $\tilde{U}$ and $\tilde{V}$ are now orthogonal, and they can be made to have unit norm by normalizing each column. As these extensions exhibit the same symmetries of $U$ and $V$, respectively, analogous arguments to the proof of inseparability of $U$ and $V$ by EPNN (Theorem 2) will apply to this new pair $\tilde{U}, \tilde{V}$. Therefore, EPNN cannot distinguish $\tilde{U}$ and $\tilde{V}$.

## A.3 Proofs for implications for real-world GNNs

**Proposition 3.** Vanilla OGE-Aug [53] is incomplete over graphs with a simple spectrum.

*Proof.* The method proposed by Zhou et al. [53] consists of a permutation equivariant and orthogonal invariant function. We will show that a counterexample by [30] also applies to this network.

Vanilla PGE-Aug relies on a permutation-equivariant and orthogonal invariant set encoding to process each eigenspace separately. Werevisit their separate definitions and theorems:

**Definition 4** (O(p)-invariant universal representation [53]). *Let $f : \bigcup_{n=0}^{\infty} \mathbb{R}^{n \times p} \to \bigcup_{n=0}^{\infty} \mathbb{R}^n$. Given an input $V \in \mathbb{R}^{n \times p}$, $f$ outputs a vector $f(V) \in \mathbb{R}^n$. The function $f$ is said to be an $O(p)$-invariant universal representation if given $V, V' \in \mathbb{R}^{n \times p}$ and $P \in S_n$, the following two conditions are equivalent:*

(i) $f(V) = Pf(V')$;

(ii) $\exists Q \in O(p)$, such that $V = PV'Q$.

**Definition 5** (Universal set representation [53]). *Let $X$ be a non-empty set. A function $f : 2^X \to \mathbb{R}$ is said to be a universal set representation if $\forall X_1, X_2 \in 2^X$, $f(X_1) = f(X_2)$ if and only if the two sets $X_1$ and $X_2$ are equal.*

**Proposition 3.5** (Zhou et al. [53]) For each $p = 1, 2, \ldots$, let $f_p$ be an $O(p)$-invariant universal representation function. Further let $g : 2^{\mathbb{R}^3} \to \mathbb{R}$ be a universal set representation. Then the following function

$$r(G, X_G) = \text{GNN}\left(A_G, \text{concat}\left[X_G, g\left(\{\text{concat}[\mu_j \mathbf{1}_n, \lambda_j \mathbf{1}_n, f_{\mu_j}(V_j)]\}_{j=1}^K\right)\right]\right) \tag{8}$$

is a universal representation. Here $n = |V(G)|$, $((\lambda_1, \mu_1), \ldots, (\lambda_K, \mu_K))$ is the spectrum of $G$, and $V_j \in \mathbb{R}^{n \times \mu_j}$ are the $\mu_j$ mutually orthogonal normalized eigenvectors of $L_G$ corresponding to $\lambda_j$. We denote $\mathbf{1}_n$ an all-1 vector of shape $n \times 1$. GNN is a maximally expressive MPNN.

Then Zhou et al. [53] propose the following graph neural network:

**Definition 3.6** (Vanilla OGE-Aug). Let $f_p$ be an $O(p)$-invariant universal representation, for each $p = 1, 2, \ldots$, and $g : 2^{\mathbb{R}^3} \to \mathbb{R}$ be a universal set representation. Define $Z : \mathcal{G} \to \bigcup_{n=1}^\infty \mathbb{R}^n$ as

$$Z(G) = g\left(\left\{\text{concat}\left[\mu_j \mathbf{1}_{|V(G)|}, \lambda_j \mathbf{1}_{|V(G)|}, f_{\mu_j}(V_j)\right]\right\}_{j=1}^K\right), \tag{5}$$

In which the notations follow Proposition 3.5. For $G \in \mathcal{G}$, $Z(G)$ is called a **vanilla orthogonal group equivariant augmentation**, or **Vanilla OGE-Aug** on $G$.

We will show that architectures of the form of Proposition 3.5 and specifically Vanilla OGE-Aug are incomplete on simple spectrum graphs, contradicting the claim in Proposition 3.5 that such a representation is universal.

Consider the point clouds proposed by Ma et al. [30]:

$$U_1 = [u_{11}, u_{12}] = \begin{pmatrix} 1 & -1 & 1 & -1 \\ 2 & 3 & 4 & 5 \end{pmatrix}^\top, \tag{9}$$

$$U_2 = [u_{21}, u_{22}] = \begin{pmatrix} -1 & 1 & 1 & -1 \\ 2 & 3 & 4 & 5 \end{pmatrix}^\top. \tag{10}$$

Suppose the first column eigenvector of $U_1$ and $U_2$ corresponds to eigenvalue $\lambda_1 = 1$, the second column eigenvector of $U_1$ and $U_2$ corresponds to eigenvalue $\lambda_2 = 2$, and other eigenvectors not shown corresponds to eigenvalue $0$ (so we safely ignore them). Then the Laplacian matrices corresponding to $U_1$ and $U_2$ are:

$$L_1 = \lambda_1 u_{11} u_{11}^\top + \lambda_2 u_{12} u_{12}^\top = \begin{pmatrix} 9 & 11 & 17 & 19 \\ 11 & 19 & 23 & 31 \\ 17 & 23 & 33 & 39 \\ 19 & 31 & 39 & 51 \end{pmatrix}, \tag{11}$$

$$L_2 = \lambda_1 u_{21} u_{21}^\top + \lambda_2 u_{22} u_{22}^\top = \begin{pmatrix} 9 & 11 & 15 & 21 \\ 11 & 19 & 25 & 29 \\ 15 & 25 & 33 & 39 \\ 21 & 29 & 39 & 51 \end{pmatrix}. \tag{12}$$

We will now demonstrate the model in Proposition 3.5 will be unable to distinguish $U_1$ and $U_@$, regardless of the choice of the GNN.

First, consider an arbitrary $O(1)-$invariant representation $f : \mathbb{R}^n \to \mathbb{R}^n$. We will show that $f(U_1)$ and $f(U_2)$ are identical.

By the permutation equivariance and $O(1)$ invariance:

$$f(u_{11}) = f(-u_{11}) = f(P_{11} u_{11}) = P_{11} f(u_{11}) \tag{13}$$

where $P_{11}$ is any permutation that satisfies $P_{11}u_{11} = -u_{11}$. Therefore $P_{11}$ can be chosen to be $\sigma_1 \triangleq (1\ 2)\,(3\ 4)$ or $\sigma_2 \triangleq (1\ 4)\,(2\ 3)$.

By Equation 13, and since equality is a transitive relation, it holds that $f(u_{11})(i) = f(u_{11})(j)$ for any $i$ and $j$ in the same orbit under the group $< \sigma_1, \sigma_2 >$, the group generated by $\sigma_1$ and $\sigma_2$. It is easy to check any pair $(i,j) \in \{1,2,3,4\}^2$ can be transposed under a group element in the generated group. Therefore, $f(u_{11})$ is a constant function. Analogous arguments yield $f(u_{21})$ is also constant.

Note that for $P_{12} \triangleq (1\ 2)(3\ 4)$, it holds that

$$f(u_{11}) \underbrace{=}_{f(u_{11})\text{ is constant}} Pf(u_{11}) \underbrace{=}_{\text{perm. equivariance}} f(Pu_{11}) = f(u_{21})$$

Therefore, $f(u_{11}) = f(u_{21})$. Moreover, the second eigenvectors, $u_{12}$ and $u_{22}$ of $U_1$ and $U_2$, respectively, are identical therefore clearly $f(u_{12}) = f(u_{22})$.

This analysis naturally extends to a proper eigendecomposition (orthonormal eigenvectors of a graph as proposed by Ma et al. [30] in the proof of their Corollary 3.5 [30].

Therefore, as any universal, invariant set representation is the same on both $U_1$ and $U_2$, the input to the network will be identical per its definition, and thus for their corresponding graphs $G_1$ and $G_2$ and identical node features $X_{G_1}$ and $X_{G_2}$, respectively it holds that

$$r(G_1, X_{G_1}) = r(G_2, X_{G_2})$$

yet $G_1$ and $G_2$ are non-isomorphic, thus Vanilla OGE-Aug is incomplete.

$\square$

## A.4 Proof for equiEPNN strictly more expressive

**Corollary 1.** *equiEPNN (see Section 3.3) can separate $U$ and $V$ after 2 iterations. Thus equiEPNN is strictly stronger than EPNN.*

*Proof.* We show that after a single iteration, the equivariant update step can yield new matrices $U^{(t)}, V^{(t)}, t = 1$ which have no zeros. We can choose the update function $\text{UPDATE}_{(1,2)}$ such that $\text{UPDATE}_{(1,2)}(v_5 \odot v_5, v_1 \odot v_1, v_5 \odot v_1) \triangleq (1,1,0,0,0,0,0)$ and for all other values we define it as $\vec{0}$.

After a single iteration $U^{(1)}$ and $V^{(1)}$ will be

$$U^{(1)T} = \begin{pmatrix} z_0 & z_1 & z_2 & z_3 & z_0 & 0_2 & 0_2 & 0_2 & z_0 & z_1 & z_2 & z_3 \\ z_0 & z_1 & z_2 & z_3 & z_0 & z_1 & z_2 & z_3 & 0_2 & 0_2 & 0_2 & 0_2 \\ 0_2 & 0_2 & 0_2 & 0_2 & z_0 & z_1 & z_3 & z_2 & z_0 & z_2 & z_1 & z_3 \end{pmatrix}$$

$$V^{(1)T} = \begin{pmatrix} z_0 & z_1 & z_2 & z_3 & z_0 & 0_2 & 0_2 & 0_2 & z_0 & z_1 & z_2 & z_3 \\ z_0 & z_1 & z_2 & z_3 & z_0 & z_1 & z_2 & z_3 & 0_2 & 0_2 & 0_2 & 0_2 \\ 0_2 & 0_2 & 0_2 & 0_2 & z_1 & z_0 & z_2 & z_3 & z_2 & z_0 & z_3 & z_1 \end{pmatrix}$$

Since there exists a column (the fifth column) such that all its entries are non-zero in both $U^{(1)T}$ and $V^{(1)T}$, from Theorem 2, we know that a single iteration of EPNN, and hence also of equiEPNN, can separate $U^{(1)T}, V^{(1)T}$. In conclusion, two iterations of equiEPNN are sufficient for separation. $\square$

## A.5 Proof of Incompleteness of Equivariant EPNN

The purpose of this section is to show that equivariant EPNN is also not complete on simple spectrum graphs. To show this, we will construct a counter-example of a pair $X, Y$ which are not isomorphic with respect to the joint action of permutations and sign multiplications, and yet cannot be distinguished by equiEPNN. We note that the columns of $X, Y$ are not orthonormal, and they do have automorphisms.

For $X, Y \in \mathbb{R}^{n \times K}$, we will say that $X \equiv Y$, if there is some permutation matrix $P$ such that $PX = Y$. We will say that $s \in \{-1, 1\}^K$ is an **isomorphism** between $X$ and $Y$, if $X \operatorname{diag}(s) \equiv Y$. Here $\operatorname{diag}(s)$ is the $K \times K$ diagonal matrix with $s$ on the diagonal. An **automorphism** of $X$ is an isomorphism from $X$ to $X$.

As a first step to construct our counter example, we consider the subgroup $H \leq \{-1, 1\}^3$ defined by
$$H = \{s \in \{-1, 1\}^3 \mid s_1 \cdot s_2 \cdot s_3 = 1\}.$$
Let $T$ be the $4 \times 3$ matrix whose rows are the four elements of $H$, namely
$$T = \begin{pmatrix} 1 & 1 & 1 \\ 1 & -1 & -1 \\ -1 & -1 & 1 \\ -1 & 1 & -1 \end{pmatrix}$$
Note that $\operatorname{Aut}(T) = H$ due to $H$ having a group structure.

Next, we build the matrix $X$ to consist of four different copies of $T$. Each copy will be not a $4 \times 3$ but a $4 \times 6$ matrix, where three of the columns are the columns of $T$, and the rest are zero columns. Moreover, any two copies of $T$ will only have one non-zero column in common.

To do this, we choose four index sets in $\{1, 2, \ldots, 6\}$, who have this intersection pattern, namely
$$I_1 = \{1, 2, 3\}, \quad I_2 = \{3, 4, 5\}, \quad I_3 = \{2, 4, 6\}, \quad I_4 = \{1, 5, 6\}.$$
One can verify that indeed $|I_j \cap I_k| = 1$ for all $j \neq k$. We then define the matrix $T[I_j]$ to be the $4 \times 6$ matrix as described previously. For example
$$T[I_2] = \begin{pmatrix} 0 & 0 & 1 & 1 & 1 & 0 \\ 0 & 0 & 1 & -1 & -1 & 0 \\ 0 & 0 & -1 & -1 & 1 & 0 \\ 0 & 0 & -1 & 1 & -1 & 0 \end{pmatrix}$$
We define $X$ to be the block matrix
$$X = \begin{pmatrix} T[I_1] \\ T[I_2] \\ T[I_3] \\ T[I_4] \end{pmatrix} \in \mathbb{R}^{16 \times 6}$$

or explicitly

$$X = \left( \begin{array}{cccccc} 1 & 1 & 1 & 0 & 0 & 0 \\ 1 & -1 & -1 & 0 & 0 & 0 \\ -1 & -1 & 1 & 0 & 0 & 0 \\ -1 & 1 & -1 & 0 & 0 & 0 \\ \hline 0 & 0 & 1 & 1 & 1 & 0 \\ 0 & 0 & 1 & -1 & -1 & 0 \\ 0 & 0 & -1 & -1 & 1 & 0 \\ 0 & 0 & -1 & 1 & -1 & 0 \\ \hline 0 & 1 & 0 & 1 & 0 & 1 \\ 0 & 1 & 0 & -1 & 0 & -1 \\ 0 & -1 & 0 & -1 & 0 & 1 \\ 0 & -1 & 0 & 1 & 0 & -1 \\ \hline 1 & 0 & 0 & 0 & 1 & 1 \\ 1 & 0 & 0 & 0 & -1 & -1 \\ -1 & 0 & 0 & 0 & -1 & 1 \\ -1 & 0 & 0 & 0 & 1 & -1 \end{array} \right)$$

We define $Y$ similarly, but we elementwise multiply the rows of $T[I_1]$ by the sign vector
$$q = [-1, 1, 1, 1, 1, 1]$$
to obtain
$$Y = \begin{pmatrix} T[I_1]\operatorname{diag}(q) \\ T[I_2] \\ T[I_3] \\ T[I_4] \end{pmatrix} \in \mathbb{R}^{16 \times 6}.$$
This is our counterexample. We claim/

**Theorem 4.** *(Incompleteness of Equivariant EPNN) There exist $X, Y \in \mathbb{R}^{16 \times 6}$ such that the following statements hold:*

1. *$X$ and $Y$ are not isomorphic under the group action of $S_{16} \times \{-1, 1\}^6$.*

2. *Equivariant EPNN cannot separate $X$ and $Y$ after any number of iterations.*

*Therefore, Equivariant EPNN is incomplete on simple spectrum graphs.*

*Proof.* Remark: $X, Y$ **are not isomorphic.** By considering the zero patterns of $X$ and $Y$, one sees that if $s \in \{-1, 1\}^6$ is an isomorphism mapping $X$ to $Y$, then $s$ satisfies $P^T T[I_1] \text{diag}(s) = T[I_1] \text{diag}(q)$, for some permutation matrix $P$ (acting on the rows of $T[I_1]$), and $s$ must also define an automorphism of $T[I_j]$ for $j = 2, 3, 4$. Since each $T[I_j]$'s rows (padded with zeros) form a group, its only automorphisms are elementwise multiplications of its rows by its group elements, which implies $\prod_{i \in I_j} s_i = 1$. From these three automorphism conditions, we deduce:

$$s_3 \cdot s_4 \cdot s_5 = 1$$
$$s_2 \cdot s_4 \cdot s_6 = 1$$
$$s_1 \cdot s_5 \cdot s_6 = 1$$

Multiplying these three equations with each other we deduce that

$$s_1 \cdot s_2 \cdot s_3 = 1.$$

Now, if this holds, then $s$ cannot satisfy $P^T T[I_1] \text{diag}(s) = T[I_1] \text{diag}(q)$ because the product of the first three entries of any row of $P^T T[I_1] \text{diag}(s)$ is 1, while the product of the first three entries of any row of $T[I_1] \text{diag}(q)$ is $-1$.

### $X$ and $Y$ **cannot be separated by equiEPNN**

We prove by induction that for any number of layers in an equiEPNN, the hidden states for nodes within the same partition $B_k$ are identical, and this holds for both graph structures $X$ and $Y$. This equivalence prevents the network from separating them.

We introduce useful definitions:

**Definition 6** (Block Structure and Neighborhoods). *We partition the $n = 16$ nodes (rows) into 4 disjoint blocks $B_k$ for $k = 1, \dots, 4$ (e.g., $B_1 = \{1, \dots, 4\}$, $B_2 = \{5, \dots, 8\}$, etc.). The $4 \times 6$ matrix of initial equivariant features for block $B_k$ is $B_k^{(0)} \triangleq X[B_k, :] = T[I_k]$. The non-zero feature indices for this block are $I_k$. For a node $i \in B_k$, we define its neighbors: $\mathcal{N}_{intra}(i) \triangleq B_k$ and $\mathcal{N}_{inter}(i) \triangleq \{1, \dots, n\} \setminus B_k$.*

**Definition 7** (Invariant Node Neighborhood). *The message from a neighbor $j$ to a node $i$ of $X$ is a tuple containing the neighbor's invariant features and an invariant computed from their equivariant features, $(h_j^{(l)}, x_i^{(l)} \odot x_j^{(l)})$. The Invariant Node Neighborhood of a node $i$ at layer $l$ is the multiset of invariant features $\mathcal{I}_i^{(l)} \triangleq \mathcal{I}_{i,intra}^{(l)} \cup \mathcal{I}_{i,inter}^{(l)}$, where*

- $\mathcal{I}_{i,intra}^{(l)} = \{(h_j^{(l)}, x_i^{(l)} \odot x_j^{(l)}) \mid j \in \mathcal{N}_{intra}(i)\}$

- $\mathcal{I}_{i,inter}^{(l)} = \{(h_j^{(l)}, x_i^{(l)} \odot x_j^{(l)}) \mid j \in \mathcal{N}_{inter}(i)\}$

*where $\{\cdot\}$ denotes a multi-set. The update rule combines the node's own invariant state $h_i^{(l)}$ with aggregations of the messages from its neighborhoods:*

$$h_i^{(l+1)} = \phi_h(h_i^{(l)}, AGG(\mathcal{I}_i^{(l)}))$$

*where AGG is a permutation-invariant aggregation function (e.g., sum or mean).*

**Definition 8** (Equivariant Node Neighborhood). *The Equivariant Node Neighborhood of a node $i$ at layer $l$ is defined by $\mathcal{E}_i^{(l)} \triangleq \mathcal{E}_{i,intra}^{(l)} \cup \mathcal{E}_{i,inter}^{(l)}$, where*

- *Intra-block Neighborhood $\mathcal{E}_{i,intra}^{(l)} = \{\phi_v(h_i^{(l)}, h_j^{(l)}, x_i^{(l)}, x_j^{(l)}) \odot x_j^{(l)} \mid j \in \mathcal{N}_{intra}(i)\}$*

- *Inter-block Neighborhood $\mathcal{E}_{i,inter}^{(l)} = \{\phi_v(h_i^{(l)}, h_j^{(l)}, x_i^{(l)}, x_j^{(l)}) \odot x_j^{(l)} \mid j \in \mathcal{N}_{inter}(i)\}$*

*where $\{\cdot\}$ denotes a multi-set. Also, define the messages arriving to a node $i \in B_p$ from a different block $B_k$ ($k \neq p$) at layer $l$ by $\mathcal{E}_{i,k}^{(l)} = \{\phi_v(h_i^{(l)}, h_j^{(l)}, x_i^{(l)}, x_j^{(l)}) \odot x_j^{(l)} \mid j \in B_k\}$.*

*The equivariant feature is updated by summing over both neighborhoods:*

$$x_i^{(l+1)} = x_i^{(l)} + \sum_{m \in \mathcal{E}_{i,intra}^{(l)}} m + \sum_{m \in \mathcal{E}_{i,inter}^{(l)}} m$$

*Proof outline:*

1. We show that the blocks of X and Y are a particular case of a generalized block structure.

2. We analyze the mechanics of equiEPNN when processing these generalized X and Y to prove that the invariant node neighborhoods of corresponding nodes in X and Y are equivalent. This is the base of our induction.

3. We show that the equivariant update step maintains this generalized block structure for both $X$ and $Y$. The equivariant update maintaining the generalized block pattern of $X$ and $Y$ is the induction step of the proof.

4. Since an equivariant update maintains the generalized block structure of $X$ and $Y$, and the subsequent invariant node neighborhoods of corresponding points in generalized $X$ and $Y$ are identical, by the base of induction, equiEPNN will output the same readout for both $X$ and $Y$ after arbitrarily many refinement iterations (the hidden states are equivalent as multisets for both point clouds).

**Base Case (Generalized Block Pattern and Invariant Update)**

**Generalized Block Pattern** The initial invariant features $h_i^{(0)} = x_i^{(0)} \odot x_i^{(0)}$ are identical for all $i \in B_k$, as they equal the indicator vector for the partition $I_k$. We consider a generalized case, where the initial equivariant features for block $B_k$ (with non-zero columns $I_k$) form a matrix $B_k^{(0)}$ where $B_k^{(0)}[:, I_k]$ is:

$$B_k^{(0)}[:, I_k] = \begin{bmatrix} s_{k,1}^{(0)} & s_{k,2}^{(0)} & s_{k,3}^{(0)} \\ -s_{k,1}^{(0)} & -s_{k,2}^{(0)} & s_{k,3}^{(0)} \\ s_{k,1}^{(0)} & -s_{k,2}^{(0)} & -s_{k,3}^{(0)} \\ -s_{k,1}^{(0)} & s_{k,2}^{(0)} & -s_{k,3}^{(0)} \end{bmatrix}$$

In our counterexample $X$, the scalars $s_{k,j} \equiv 1$ for all $k, j$. For $Y$, $s_{1,1} = -1$ (from block $k = 1$, column $j = 1$) and all other $s_{k,j} \equiv 1$. We consider this generalized case because we will show that after an equivariant aggregation, this will be the format of the blocks. These are called generalized $X$ and $Y$ with a single scalar choice defining them, as the generalized $Y$ is equivalent to generalized $X$ up to a negation first row of the first block of $X$. We refer to these generalized $X$ and $Y$ as simply $X$ and $Y$ in the remainder of the proof.

These initial hidden states $h_i^{(0)}$ are identical for both point clouds $X$ and $Y$, due to the invariance of squaring to sign changes. Additionally, $h_i^{(0)} = h_j^{(0)}$ for $i, j \in B_k$ and $h_i^{(0)} \neq h_j^{(0)}$ for $j \notin B_k$, due to the unique sparsity pattern of each block. This completes the first step of the outline. We now proceed to the second step of the outline, where we prove that an invariant update maintains the equivalence of the hidden states within each block.

**Invariant Update** We formally define the aggregation steps for a node $i \in B_k$ at layer $l$ by splitting our analysis of its neighborhood into intra-block neighbors $\mathcal{N}_{intra}(i)$ and inter-block neighbors $\mathcal{N}_{inter}(i)$. For any two nodes $i, j \in B_k$, we show their invariant neighborhoods yield identical aggregations.

- **Intra-block:** A message from a neighbor $m \in B_k$ is $(h_m^{(l)}, x_i^{(l)} \odot x_m^{(l)})$. By the inductive hypothesis, $h_m^{(l)}$ is constant for all $m \in B_k$. The set of vectors $\{x_m^{(l)} \mid m \in B_k\}$ forms a group under the Hadamard product (up to scalar multiples). By the group closure property, the multiset of products $\{x_i^{(l)} \odot x_m^{(l)} \mid m \in \mathcal{N}_{\text{intra}}(i)\}$ is simply a permutation of $\{x_j^{(l)} \odot x_m^{(l)} \mid m \in \mathcal{N}_{\text{intra}}(j)\}$ for any $i, j \in B_k$. Therefore, any permutation-invariant aggregation over $\mathcal{I}_{i,\text{intra}}^{(l)}$ and $\mathcal{I}_{j,\text{intra}}^{(l)}$ is identical.

- **Inter-block:** The graph is constructed such that for any $k \neq p$, $|I_k \cap I_p| = 1$. The inter-block neighborhood for a node in $B_k$ consists of nodes from the other three blocks. Consider a neighbor $m \in B_p$. The product $x_i^{(l)} \odot x_m^{(l)}$ is non-zero only at the single index $j = I_k \cap I_p$. Due to this structure, the resulting multiset of invariants from block $B_p$ is of the form $\{\alpha_j \mathbf{e}_j, \alpha_j \mathbf{e}_j, -\alpha_j \mathbf{e}_j, -\alpha_j \mathbf{e}_j\}$ (where $\mathbf{e}_j$ is the standard basis vector and $\alpha_j$ is some scalar), which is identical for all $i \in B_k$.

Since both neighborhood aggregations are identical, and $h_i^{(l)} = h_j^{(l)}$ for $i, j \in B_k$, the update yields $h_i^{(l+1)} = h_j^{(l+1)}$ for both $X$ and $Y$.

**Inductive Step**

Assume at layer $l$, for any partition $B_k$, $h_i^{(l)} = h_j^{(l)}$ for all $i, j \in B_k$, and the equivariant feature matrix $B_k^{(l)} \triangleq X[B_k, :]^{(l)}$ maintains the scaled pattern structure.

**Equivariant Update** The update for the equivariant features $x_i^{(l+1)}$ combines the original features $x_i^{(l)}$ with aggregations from intra-block and inter-block neighbors.

**Intra-block Aggregation:** The aggregation of messages within a block $B_k$ can be compactly expressed via summation and Hadamard products. The message function $\phi_v$ produces scalar weights for each interaction. Since the invariant features $h^{(l)}$ are constant within the block, these weights depend only on the structural relationship between nodes $i$ and $j$. Due to the graph's symmetries, there are only four unique interaction types within a block, resulting in four learned scalar vectors, with scalar dimension weights in each feature dimension in $\mathbb{R}^K$. Since only $I_k$ are the indices with non-zero features, we focus on their aggregation, and the rest of the inputs along other feature dimensions will be aggregated to 0, therefore we denote by $a, b, c, d \in \mathbb{R}^3$ the reduction into the feature indices in $I_k$. These form a symmetric weight matrix (in the node dimension) we denote by

$$\Phi_k^{(l)} = \begin{bmatrix} a & b & c & d \\ b & a & d & c \\ c & d & a & b \\ d & c & b & a \end{bmatrix} \in \mathbb{R}^{4 \times 4 \times 3}$$

This operation, which we denote by $\star$, scales the columns of the feature matrix $B_k^{(l)}$ while preserving their sign-pattern structure:

$$\Phi_k^{(l)} \star B_k^{(l)}[:, I_k] = \begin{bmatrix} a \odot B_k^{(l)}[1, I_k] + b \odot B_k^{(l)}[2, I_k] + c \odot B_k^{(l)}[3, I_k] + d \odot B_k^{(l)}[4, I_k] \\ b \odot B_k^{(l)}[1, I_k] + a \odot B_k^{(l)}[2, I_k] + d \odot B_k^{(l)}[3, I_k] + c \odot B_k^{(l)}[4, I_k] \\ c \odot B_k^{(l)}[1, I_k] + d \odot B_k^{(l)}[2, I_k] + a \odot B_k^{(l)}[3, I_k] + b \odot B_k^{(l)}[4, I_k] \\ d \odot B_k^{(l)}[1, I_k] + c \odot B_k^{(l)}[2, I_k] + b \odot B_k^{(l)}[3, I_k] + a \odot B_k^{(l)}[4, I_k] \end{bmatrix} \in \mathbb{R}^{4 \times 3} \tag{14}$$

$$= \begin{bmatrix} \alpha & \beta & \gamma \\ -\alpha & -\beta & \gamma \\ \alpha & -\beta & -\gamma \\ -\alpha & \beta & -\gamma \end{bmatrix} \tag{15}$$

where the entries of the resulting matrix (in the $I_k$ columns) are denoted by the scalars $\alpha, \beta, \gamma$. These scalars are the result of applying the learned weights $a, b, c, d$ (which are vectors) to the corresponding

columns of $B_k^{(l)}$. Specifically, they are defined as:

$$\alpha = s_{k,1}^{(l)}(a_{i_1} - b_{i_1} + c_{i_1} - d_{i_1})$$

$$\beta = s_{k,2}^{(l)}(a_{i_2} - b_{i_2} - c_{i_2} + d_{i_2})$$

$$\gamma = s_{k,3}^{(l)}(a_{i_3} + b_{i_3} - c_{i_3} - d_{i_3})$$

where $a_j$ is the $j$-th component of $a$, etc. and $i_1, i_2, i_3 \in I_k$. This operation preserves the fundamental sign-pattern structure of each column, merely updating its overall scaling factor.

**Inter-block Aggregation:** Let $i \in B_p$ be a node index in block $p$, and let $k \neq p$ be a different block index. Consider the equivariant node neighborhood $\mathcal{E}_{i,k}^{(l)}$. We first focus on the inter-block aggregation of $X$ and proceed to discuss that of $Y$. Consider the contribution of the equivariant message passing to the features of node $i$ from block $B_k$. There are 3 possible cases:

**Case 1: Aggregation of $\mathcal{E}_{i,k}^{(l)}$ along dimension $d = I_k \cap I_p$.** Along this single feature index $d$, the only non-zero information in the product is $x_i^{(l)}[d] \cdot x_j^{(l)}[d] \in \mathbb{R}$ for $j \in B_k$. This scalar, apart from the hidden states (which are constant within blocks), is the only structural value that determines $\phi_v(h_i^{(l)}, h_j^{(l)}, x_i^{(l)}, x_j^{(l)})$. This $\phi_v$ in turn determines the feature-wise weighing of $x_j^{(l)}$. It follows that for any node $s \in B_p$, the sum of $\mathcal{E}_{s,k}^{(l)}$ in the feature dimension $d$ precisely equals that of $i$ up to $\text{sign}(x_i^{(l)}[d] \cdot x_s^{(l)}[d])$. Because $x_s^{(l)}[d]$ follows the generalized block pattern for $B_p$, the aggregation into dimension $d$ also follows this pattern.

**Case 2: Aggregation of $\mathcal{E}_{i,k}^{(l)}$ along feature indices in $I_k \setminus I_p$.** From Case 1, $\phi_v$ is determined by the sign of the product in dimension $d$. This results in two possible weight vectors, say $\vec{a}$ and $\vec{b}$. The set of messages is

$$\mathcal{E}_{i,k}^{(l)}(X) = \{\vec{a} \odot x_{j_1}, \vec{a} \odot x_{j_2}, \vec{b} \odot x_{j_3}, \vec{b} \odot x_{j_4}\} \tag{16}$$

where $x_{j_1}, x_{j_2}$ are (w.l.o.g) the points with a positive scalar product with $x_i$ in dimension $d$, and $x_{j_3}, x_{j_4}$ are those with a negative product. By construction of $T$, the points $x_{j_1}, x_{j_2}$ satisfy $x_{j_1}(m) = -x_{j_2}(m)$ for each $m \in I_k \setminus I_p$. An analogous result holds for $x_{j_3}, x_{j_4}$. Therefore, summing all points in $\mathcal{E}_{i,k}^{(l)}$ yields zeros in feature entries $I_k \setminus I_p$.

**Case 3: Aggregation of $\mathcal{E}_{i,k}^{(l)}$ along remaining indices.** In all other indices, $\{1, 2, \ldots, 6\} \setminus (I_p \cup I_k)$, the features of $x_j$ (for $j \in B_k$) are 0. Thus, it trivially holds that after aggregating $\mathcal{E}_{i,k}^{(l)}$, the resulting vector entries in those dimensions will also be 0.

We now address the inter-block update of $Y$ in comparison with that of $X$. The only structural difference is the negated first column in block $B_1$ of $Y$. This affects aggregation for $i \in B_1$ and for $i \in B_4$ (since $I_1 \cap I_4 = \{1\}$). The sign of $x_i^{(l)}[1] \cdot x_j^{(l)}[1]$ is flipped. This means the roles of $\vec{a}$ and $\vec{b}$ are swapped. For $i \in B_1$:

$$\mathcal{E}_{i,k}^{(l)}(Y) = \{\vec{b} \odot y_{j_1}, \vec{b} \odot y_{j_2}, \vec{a} \odot y_{j_3}, \vec{a} \odot y_{j_4}\} \tag{17}$$

The sum is thus negated. This occurs only along the first column of the first block. For $i \in B_4$, the aggregation from $B_1$ is:

$$\text{AGG}(\mathcal{E}_{i,1}(X)) = \text{AGG}(\{\vec{a} \odot x_{j_1}, \vec{a} \odot x_{j_2}, \vec{b} \odot x_{j_3}, \vec{b} \odot x_{j_4}\}) \tag{18}$$

$$= \text{AGG}(\{\vec{a} \odot \mathbf{e}_1 \odot x_{j_1}, \vec{a} \odot \mathbf{e}_1 \odot x_{j_2}, \vec{b} \odot -\mathbf{e}_1 \odot x_{j_3}, \vec{b} \odot -\mathbf{e}_1 \odot x_{j_4}\}) \tag{19}$$

$$= \text{AGG}(\{\vec{b} \odot -\mathbf{e}_1 \odot y_{j_1}, \vec{b} \odot -\mathbf{e}_1 \odot y_{j_2}, \vec{a} \odot \mathbf{e}_1 \odot y_{j_3}, \vec{a} \odot \mathbf{e}_1 \odot y_{j_4}\}) \tag{20}$$

$$= \text{AGG}(\{\vec{b} \odot y_{j_1}, \vec{b} \odot y_{j_2}, \vec{a} \odot y_{j_3}, \vec{a} \odot y_{j_4}\}) = \text{AGG}(\mathcal{E}_{i,1}(Y)) \tag{21}$$

for some $\vec{a}, \vec{b} \in \mathbb{R}^6$. The equality holds. Therefore, the equivariant aggregation of $Y$ is equivalent to that of $X$, except in the first column of the first block, where it is negated. The aggregations for both $X$ and $Y$ maintain the generalized block pattern.

In conclusion of the inter-block aggregation, each $x_i \in B_k$ will be added with an equivariant feature of the form $c^{(l)} \odot x_i$ (where $c^{(l)}$ is a shared column vector), and $y_i$ will be added with $c^{(l)} \odot y_i$.

**Full Update:** The new feature matrix $B_k^{(l+1)}$ is the sum of the original features and the intra- and inter-block aggregations. This process preserves the essential column structure. The update can be expressed as:

$$B_k^{(l+1)} = (I + \Phi_k^{(l)})B_k^{(l)} + c^{(l)} \odot B_k^{(l)} \tag{22}$$

where $c^{(l)}$ is a column vector. This operation simply updates the scalar multiples of each column. For example, the sum of the original features and the intra-block aggregation (for the $I_k$ columns) results in:

$$(I + \Phi_k^{(l)})B_k^{(l)}[:, I_k] = \begin{bmatrix} s_{k,1}^{(l)} + \alpha & s_{k,2}^{(l)} + \beta & s_{k,3}^{(l)} + \gamma \\ -(s_{k,1}^{(l)} + \alpha) & -(s_{k,2}^{(l)} + \beta) & s_{k,3}^{(l)} + \gamma \\ s_{k,1}^{(l)} + \alpha & -(s_{k,2}^{(l)} + \beta) & -(s_{k,3}^{(l)} + \gamma) \\ -(s_{k,1}^{(l)} + \alpha) & s_{k,2}^{(l)} + \beta & -(s_{k,3}^{(l)} + \gamma) \end{bmatrix}$$

By Equation 22, the equivariant features remain in the generalized block form.

In conclusion, at each layer, invariant features remain uniform within partitions, and equivariant features update symmetrically. Since the representations are structurally identical (up to the $s_{1,1}$ sign flip, which is preserved) for both graphs, they are indistinguishable.

### $X$ and $Y$ can be extended to a proper eigendecomposition

To form a complete basis of 16 eigenvectors, we construct the remaining **10** orthogonal vectors, $\tilde{X} \in \mathbb{R}^{16 \times 10}$. Define the local orthogonal basis for $\mathbb{R}^4$ (along the rows of the matrix):

$$\begin{bmatrix} \mathbf{a} \\ \mathbf{b} \\ \mathbf{c} \\ \mathbf{1} \end{bmatrix} \triangleq \begin{bmatrix} 1 & -1 & 1 & -1 \\ 1 & -1 & -1 & 1 \\ 1 & 1 & -1 & -1 \\ 1 & 1 & 1 & 1 \end{bmatrix} \tag{23}$$

Let $\mathbf{0} \in \mathbb{R}^4$ be the zero vector. We construct the matrix $\tilde{X}^T \in \mathbb{R}^{10 \times 16}$ as a block matrix (where each block $\mathbf{a}, \mathbf{b}, \ldots$ is a $1 \times 4$ row vector):

$$\tilde{X}^T \triangleq \begin{bmatrix} \mathbf{a} & -\mathbf{a} & \mathbf{0} & \mathbf{0} \\ \mathbf{b} & \mathbf{0} & -\mathbf{a} & \mathbf{0} \\ \mathbf{c} & \mathbf{0} & \mathbf{0} & -\mathbf{a} \\ \mathbf{0} & \mathbf{b} & -\mathbf{b} & \mathbf{0} \\ \mathbf{0} & \mathbf{c} & \mathbf{0} & -\mathbf{b} \\ \mathbf{0} & \mathbf{0} & \mathbf{c} & -\mathbf{c} \\ \mathbf{1} & \mathbf{0} & \mathbf{0} & \mathbf{0} \\ \mathbf{0} & \mathbf{1} & \mathbf{0} & \mathbf{0} \\ \mathbf{0} & \mathbf{0} & \mathbf{1} & \mathbf{0} \\ \mathbf{0} & \mathbf{0} & \mathbf{0} & \mathbf{1} \end{bmatrix} \tag{24}$$

The rows of $\tilde{X}^T$ (columns of $\tilde{X}$) are orthogonal to each other and to the columns of $X$. Thus, after scaling, $X_{full} = [X, \tilde{X}] \in \mathbb{R}^{16 \times 16}$ forms an orthonormal basis. The block structure is maintained within the first 6 rows of $\tilde{X}^T$. An analogous proof to Part 2 shows that these do not contribute new information to the hidden states, other than their block membership. The "all-ones" vectors (last 4 rows) are constant on each block and do not pass messages between blocks. This means the invariant and equivariant aggregation remain analogous to Part 2 when processing the full matrix $X_{full} = [X, \tilde{X}]$. The hidden states only depend on the structural relations defined by $X$. Therefore, for an analogous matrix $\tilde{Y}$ (with its first row $\mathbf{a}$ replaced by $-\mathbf{a}$ to maintain orthogonality with $Y$), equiEPNN will yield the same output on $[Y, \tilde{Y}]$ and $[X, \tilde{X}]$, which form valid eigendecompositions for an equal simple spectrum. In conclusion, equiEPNN cannot separate $X$ and $Y$.

$\square$

# B Experiments

## B.1 Dataset statistics

We surveyed popular graph datasets and documented their graph spectral properties. The results are shown in Table 5. We find that the MNIST Superpixel [34] dataset is almost homogeneously composed of graphs with a simple spectrum, and we find that ($96.9\%$) of the graphs in this dataset have a full row without zeros, implying that EPNN is complete on almost all graphs.

Other datasets, such as MUTAG, ENZYMES, PROTEINS and ZINC [19, 36], contain a substantial amount of graphs with eigenvalue multiplicity 2 and 3. Despite this, the number of eigenspaces of dimensions 2 and 3 is very few per graph, averaging at around 1 per graph. On datasets with highly symmetric graphs, such as ENZYMES and PROTEINS, the graphs do not meet the sparsity condition of Theorem 2, thus EPNN will not necessarily faithfully learn the graph structure. This exemplifies the need for more expressive models that are complete on graphs with higher maximal eigenvalue multiplicity and sparse eigenvectors.

Table 5: Graph Statistics Analysis Across Different Datasets (Eigenvalue Tolerance: $10^{-4}$)

| Dataset Name | MUTAG | ENZYMES | PROTEINS | MNIST | ZINC |
|---|---|---|---|---|---|
| **Dataset Overview** | | | | | |
| Number of Graphs | 188 | 600 | 1,113 | 60,000 | 10,000 |
| **Eigenvalue Characteristics** | | | | | |
| Graphs with Distinct Eigenvalues | 41.5% (78) | 34.8% (209) | 22.1% (246) | 99.9% (59,950) | 40.7% (4,072) |
| Graphs with Multiplicity 2 Eigenvalues | 58.5% (110) | 65.2% (391) | 77.9% (867) | – | 59.3% (5,928) |
| Graphs with Multiplicity 3 Eigenvalues | 19.1% (36) | 46.2% (277) | 57.9% (644) | – | 26.2% (2,617) |
| Avg. Number of Multiplicity 2 Eigenvalues | 0.74 | 1.01 | 1.24 | – | – |
| Avg. Number of Multiplicity 3 Eigenvalues | 0.26 | 0.58 | 0.71 | – | – |
| **Eigenvector Properties** | | | | | |
| Average Ratio of Zeros | 1.67 | 4.28 | 6.39 | 0.31 | 2.52 |
| Average Number of Zeros | 31.13 | 172.93 | 817.20 | 23.16 | 61.04 |
| Graphs with a Full Row | 75.0% (141) | 35.8% (215) | 37.1% (413) | 96.9% (58,077) | 64.5% (6,447) |
| Graphs with ≤1 Zero per Eigenvector | 0.0% (0) | 6.3% (38) | 5.0% (56) | 20.2% (12,085) | 4.3% (430) |
| Graphs with Total Zeros < Vertices | 29.8% (56) | 16.3% (98) | 14.3% (159) | 89.9% (53,873) | 13.0% (1,295) |
| Graphs Meeting Any Condition | 75.0% (141) | 35.8% (215) | 37.1% (413) | 96.9% (58,077) | 64.5% (6,447) |

We surveyed the graph spectra of popular datasets to verify the need for more expressive architectures based on graph properties. We now further specify the meaning of each row of Table 5 in Table B.1.

## B.2 MNIST Superpixel

Below, in Tables 7, 8 and B.2, we list the experiment configurations and hyperparameters of the MNIST experiment.

As a toy experiment to examine the potential benefit of using equiEPNN, We implemented equiEPNN via a modification of the EGNN architecture [40] and EPNN with the same architecture, but without the eigenvector update step. For precise hyperparameter configuration, see the Appendix.

In our first experiment, we applied the proposed method on a classical task of handwritten digit classification in the MNIST dataset [24]. While almost trivial by today's standards, we use this example to verify the theoretical claims regarding expressivity on simple spectrum graphs. Our experimental setup employed both EPNN (coordinate updates disabled) and equiEPNN (coordinate updates enabled) as our models exclusively on the superpixel-based graph representation from the MNISTSuperpixels dataset. In this approach, each $28 \times 28$ image was converted into a graph where vertices correspond to superpixels and edges represent their spatial adjacency relations, each image was represented as a different graph. We tested our models with different positional encoding dimensions of $k = 3, 8, 16$ to evaluate performance across varying levels of spectral information.

For details configutions see Tanbes 7, 8, and 9.

### B.2.1 Ablation

We examined the performance of both methods on the MNIST Superpixel datasets, where the task is classification of handwritten digits. We found that equiEPNN outperforms EPNN, with the same

| Eigenvalue Characteristics | |
|---|---|
| Graphs with Distinct Eigenvalues | Graphs where all eigenvalues have multiplicity 1, meaning each eigenvalue appears exactly once in the spectrum |
| Graphs with Multiplicity 2 Eigenvalues | Graphs that have at least one eigenvalue that appears exactly twice in the spectrum |
| Graphs with Multiplicity 3 Eigenvalues | Graphs that have at least one eigenvalue that appears exactly three times in the spectrum |
| Avg. Number of Multiplicity 2 Eigenvalues | The average number of eigenbasis that have multiplicity exactly 2 |
| Avg. Number of Multiplicity 3 Eigenvalues | The average number of eigenbasis that have multiplicity exactly 3 |
| Eigenvector Properties | |
| Average Ratio of Zeros | The average proportion of zero entries found in the eigenvectors across all analyzed graphs |
| Average Number of Zeros | The average count of zero entries in the eigenvectors across all analyzed graphs |
| Graphs with a Full Row | Graphs that have at least one eigenvector with no zero entries (i.e., a "full row" in the eigenvector matrix) |
| Graphs with $\leq 1$ Zero per Eigenvector | Graphs where each eigenvector has at most one zero entry |
| Graphs with Total Zeros $<$ Vertices | Graphs where the total number of zero entries across all eigenvectors is less than the number of vertices in the graph |
| Graphs Meeting Any Condition | Graphs that satisfy at least one of the specified eigenvector properties listed above |

Table 6: Explanation of Surveyed Graph Spectral Properties

Table 7: MNIST Superpixel Experiment Configuration

| Parameter | Default Value | Description |
|---|---|---|
| k_values | [3, 8, 16] | List of k values for positional encoding dimensions |
| epochs | 30 | Number of training epochs |
| batch_size | 32 | Training batch size |
| data_dir | 'data' | Data directory path |
| device | 'cuda' | Computing device (CUDA if available) |
| early_stopping | 10 | Early stopping patience |
| output_dir | 'results' | Output directory for results |
| coord_update_options | [True, False] | Coordinate update configurations |
| random_seed | 42 | Random seed for reproducibility |

Table 8: MNIST Superpixel Network Hyperparameters

| Parameter | Default Value | Description |
|---|---|---|
| num_features | 1 | Input node features (MNIST characteristic) |
| num_classes | 10 | Output classes (MNIST digits 0-9) |
| hidden_dim | 64 | Hidden layer dimension |
| num_layers | 3 | Number of EGNN layers |
| pos_enc_dim | k | Positional encoding dimension (varies: 3, 8, 16) |
| dropout | 0.2 | Dropout rate |
| lr | 0.0005 | Learning rate |
| weight_decay | 1e-5 | Weight decay for regularization |
| norm_features | True | Normalize node features |
| norm_coords | True | Normalize coordinates |
| coord_weights_clamp | 1.0 | Clamping value for coordinate weights |
| with_pos_enc | True | Use positional encoding |
| with_proj | False | Use edge projectors |
| with_virtual_node | False | Use virtual node |
| update_coords | True/False | Coordinate update flag (both tested) |

Table 9: MNIST Superpixel Training Configuration

| Parameter | Value | Description |
|---|---|---|
| Optimizer | Adam | Optimization algorithm |
| Loss Function | NLL Loss | Negative log-likelihood loss |
| Scheduler | ReduceLROnPlateau | Learning rate scheduler |
| LR Reduction Factor | 0.5 | Factor for LR reduction |
| LR Patience | 5 | Scheduler patience |
| Min LR | 1e-6 | Minimum learning rate |
| Gradient Clipping | 1.0 | Maximum gradient norm |
| Early Stopping Patience | 10 | Training patience |

number of model parameters and hyperparameter instantiations, in the setting with few known eigenvectors. With a sufficient number of eigenvectors EPNN and equiEPNN achieve comparable results, as expected, since they are both complete on almost all graphs in MNIST Superpixel. (see k=8 and k=16 in Table 10.)

### B.3 Realizable Expressivity

The BREC [48] dataset is a graph expressivity benchmark consisting of highly symmetric graphs that high-order GNNs struggle at distinguishing, which was used by [52] to check the expressivity of EPNN. We implemented EPNN and equiEPNN via the popular EGNN [40] framework and obtained statistically identical results shown in Table 12.

### B.4 Eigenvector Canonicalization

We specify the problem setup for eigenvector canonicalization and our proposed method.

Table 10: Ablation study on MNIST Superpixel [34]. Accuracy percentage comparison with deviation over 3 trials, for different values of $K$ for EPNN and equiEPNN.

| $k$ | EPNN | EquiEPNN |
|---|---|---|
| 3 | $48.45 \pm 1.2\,\%$ | $60.95 \pm 0.9\,\%$ |
| 8 | $85.55 \pm 2.1\,\%$ | $83.56 \pm 2.5\,\%$ |
| 16 | $90.13 \pm 2.3\,\%$ | $91.37 \pm 2.2\,\%$ |

Table 11: Network Hyperparameters for Eigenvector canonicalization

| Parameter | Default Value | Description |
|---|---|---|
| num_layers | 5 | Number of message passing layers |
| emb_dim | 128 | Embedding dimension |
| in_dim | 128 | Input feature dimension |
| proj_dim | 10 | Projection dimension |
| coords_weight | 3.0 | Coordinate update weight |
| activation | relu | Activation function |
| norm | layer | Normalization type |
| aggr | sum | Aggregation function |
| residual | False | Use residual connections |
| edge_attr_dim | 20 | Edge feature dimension ($2 \times$ k_projectors) |

Table 12: Empirical performance of different GNNs on BREC (in percentages.) (Using k=3 spectral features, results of non-EPNN models from [52])

| Model | WL class | Basic | Reg | Ext | CFI | Total |
|---|---|---|---|---|---|---|
| Graphormer | SPD-WL | 26.7 | 10.0 | 41.0 | 10.0 | 19.8 |
| NGNN | SWL | 98.3 | 34.3 | 59.0 | 0 | 41.5 |
| ESAN | GSWL | 96.7 | 34.3 | 100.0 | 15.0 | 55.2 |
| PPGN | 3-WL | 100.0 | 35.7 | 100.0 | 23.0 | 58.2 |
| EPNN | EPWL | 100.0 | 35.7 | 100.0 | 4.0 | 53.5 |
| Equi-EPNN | N/A | 100.0 | 35.7 | 100.0 | 4.0 | 53.5 |

**Definition 9** (Eigenvector Canonicalization). *A canonicalization of an eigenvector $v \in \mathbb{R}^n$ is a map $\phi : \mathbb{R}^n \to \mathbb{R}^n$ such that for every $s \in O(1) \simeq \{-1, 1\}$, it holds that $\phi(sv) = \phi(v)$ and is permutation equivariant, that is for every permutation $\sigma$, $\phi(\sigma v) = \sigma \phi(v)$.*

We now define the following eigenvector canonicalization map via the steps

1. For given eigenvectors $V \in \mathbb{R}^{n \times k}$ corresponding to distinct eigenvalues, we run equiEPNN for $T$ iterations, to obtain the equivariant output $V^{(T)}$

2. We sum over the columns to obtain a matrix $S = \mathrm{diag}(s_1, s_2, \ldots, s_k)$ where $s_i \triangleq \mathrm{sign}(\sum_{j=1}^n V^{(T)}(i, j)) \in \{-1, +1\}$.

3. Canonicalize the eigenvectors via $SV$.

This defines an eigenvector canonicalization map $\psi : \mathbb{R}^{n \times k} \to \mathbb{R}^{n \times k}$ where $\psi(V) = SV$ for the $S(V)$ defined above. This map is naturally permutation equivariant, and it is easy to check that it is sign invariant.

As this maps canonicalized the original eigenvectors via aggregating global graph information that depends on the entire graph eigendecomposition and not each eigenvector separately, we obtain a map that practically achieves perfect canonicalization on ZINC [19].

See Tables 11 and 13 for experiment configurations.

Table 13: Eigenvector Canonicalization Configuration

| Parameter | Default Value | Description |
|---|---|---|
| subset_size | 100 | Number of ZINC graphs to test |
| k_projectors | 10 | Number of top eigenvalue projectors to use |
| num_workers | 4 | Number of workers for data loading |
| device | CUDA/CPU | Computing device (CUDA if available) |
| precision | float64 | Default tensor precision |

# C   Further Related Work

## C.1   Expressive Power and the Weisfeiler-Lehman Hierarchy

The expressive power of GNNs is commonly evaluated via the Weisfeiler-Lehman (WL) test, with standard Message Passing Neural Networks (MPNNs) being upper-bounded by the 1-WL test [49, 35]. This has motivated the development of more powerful models aligned with higher-order k-WL tests [32]. The WL hierarchy and its variants have been clarified in tutorials by [17, 37]. Other works have moved beyond the binary isomorphism objective to develop more continuous, fine-grained measures of expressivity based on graphons and tree distances [7]. Our work diverges from these combinatorial frameworks by proposing a hierarchy based on eigenvalue multiplicity, a natural concept in spectral graph theory. We demonstrate that even SGNNs considered powerful in the WL hierarchy (EPNN) can fail on spectrally-defined graph classes, revealing limitations not captured by combinatorial tests.

## C.2   Higher-Order and Subgraph GNNs

To overcome the 1-WL barrier, a prominent line of research has focused on architectures that process higher-order structures. Subgraph GNNs, which represent a graph as an equivariant collection of its subgraphs, have proven to be a particularly powerful paradigm [6]. A significant challenge has been the computational complexity of these models. Recent work by [3] introduces a flexible and scalable framework for Subgraph GNNs using graph products and coarsening to manage complexity. This line of research, including work by [11], has also explored novel methods to boost expressivity by leveraging high-order derivatives of a base GNN model, drawing deep connections between this calculus-based approach and the WL hierarchy. Our work provides a complementary perspective by showing that even highly expressive architectural paradigms can have fundamental blind spots, such as the inability to distinguish certain graphs with simple spectra.

## C.3   Spectral GNNs and Universality

Spectral GNNs define graph convolutions via spectral filters. Early work improved filter expressivity by moving from polynomials to complex rational functions, as in CayleyNets [25]. A key theoretical result from [47] established that linear spectral GNNs can achieve universal approximation on graphs with a simple spectrum. However, this universality relies on a crucial assumption: the use of a randomly sampled, non-equivariant node signal. This setting is distinct from the standard GNN expressivity analysis, which assumes permutation-equivariant operations on graph structure. Our work investigates the expressivity of permutation-equivariant SGNNs, such as EPNN, under the same simple spectrum condition. We prove that, in this more standard setting, these models are fundamentally incomplete. We construct explicit counterexamples of non-isomorphic graphs with simple spectra that EPNN cannot distinguish, revealing a critical limitation that was not apparent from prior analyses.

## C.4   Equivariant Design and Generalization

A core principle in modern GNN theory is designing architectures that respect the symmetries of graph data, i.e., permutation invariance and equivariance [46]. This has led to principled methods for handling spectral features, such as the sign and basis ambiguities of eigenvectors. Models like SignNet and BasisNet are designed to be invariant to these symmetries by processing eigenspaces independently [27]. Work by [22] has analyzed the implicit bias of such equivariant networks, showing that gradient descent favors solutions with specific structural properties in the Fourier domain. Other work has explored probabilistic frameworks for breaking symmetries when necessary [23]. Our work builds on these principles; we show that even a principled equivariant architecture like EPNN is incomplete, and our proposed solution, equiEPNN, is directly inspired by equivariant network designs.

## C.5   Unified Theories and GNN Limitations

One recent research direction is to move towards a more holistic understanding of GNNs by connecting expressivity, generalization, and universality. Work by [39] proposes a unified framework using pseudometrics based on optimal transport to derive both universal approximation theorems and

generalization bounds for MPNNs on attributed graphs. Concurrently, critical work has highlighted the practical limitations of GNNs. For instance, [4] demonstrated that GNNs can 'overfit' the graph structure, using it even when it is detrimental to the task. This suggests that theoretical expressivity does not automatically translate to better performance. Our paper contributes to this line of inquiry by identifying a novel and unexpected failure mode for a class of GNNs that are already considered highly expressive. This reinforces the notion that expressivity is not monolithic and that different architectures have distinct failure modes.

