# OpenReview forum: "Spectral Graph Neural Networks are Incomplete on Graphs with a Simple Spectrum"
_NeurIPS.cc/2025/Conference — NeurIPS 2025 spotlight_

### Official Review · Reviewer_AgJS · 2025-06-06

**Clarity:** 3
**Significance:** 3
**Originality:** 3
**Rating:** 5
**Confidence:** 3

**Summary:**

This paper studies the limitations of spectral GNNs on graphs with a simple spectrum (distinct eigenvalues). It shows that Eigenspace Projection GNNs (EPNNs) are not expressive enough to distinguish such graphs. To address this, the authors propose equiEPNN, a sign-equivariant variant that improves expressivity. They provide theoretical proofs and empirical results showing that equiEPNN outperforms EPNN in distinguishing graph structures and canonicalizing eigenvectors.

**Questions:**

- The proposed equiEPNN assumes access to full eigenvector decomposition, which can be computationally expensive for large graphs. How does the method scale in practice? Is it feasible for datasets like ogbg-ppa large molecule graphs?
- The paper presents completeness results for EPNN under specific sparsity and automorphism assumptions. How restrictive are these in practice?
- The proposed method relies on simple spectrum graphs and the sparsity condition outlined in Theorem 2. However, according to the Statistics Analysis in Table 1, most real-world datasets appear to not satisfy both conditions simultaneously.
- Would it be possible to provide more experimental results on datasets that fully satisfy the above conditions? Additionally, can the proposed method achieve comparable performance on datasets like ENZYMES and PROTEINS, which do not strictly meet the required assumptions?
- The paper primarily focuses on graph-level tasks and the expressivity of models in distinguishing whole graphs. Have you considered how the proposed equiEPNN performs on node-level tasks, such as node classification or node clustering? Given that spectral methods often rely on global eigenvectors, how would equiEPNN handle localized, node-specific predictions in practice?

**Ethical Concerns:**

["NO or VERY MINOR ethics concerns only"]

**Final Justification:**

The authors conducted a lot of experiments to verify the proposed method from various perspectives. I recommend Accept, as the authors have convincingly addressed the major concerns, and the paper contributes a solid and practical approach.

**Limitations:**

Yes

**Quality:**

3

**Strengths And Weaknesses:**

**Strengths:**
- The paper presents strong theoretical contributions, rigorously proving the incompleteness of spectral GNNs such as EPNN on graphs with a simple spectrum.
- The proposed equiEPNN method is well-motivated and theoretically grounded, offering a principled improvement through sign-equivariant feature updates.
- The experimental section, although limited in scope, supports the theoretical claims—especially with ablation studies on MNIST Superpixel and eigenvector canonicalization on ZINC.
- The paper is well-structured, with a clear flow from problem statement to proposed solution and evaluation. The definitions, assumptions, and theoretical claims are clearly stated, and the key proofs are presented in detail.


**Weaknesses:**
- The empirical evaluation is somewhat narrow in scope and does not include more challenging benchmarks, such as large-scale graph datasets like ogbg-ppa.
- Some derivations (e.g., the canonicalization pipeline in equiEPNN) could benefit from further implementation or ablation details.
- The practical implications for broader graph learning tasks (e.g., node classification, link prediction) are not fully explored.
- The approach assumes access to full eigenvector information, which may be computationally prohibitive on large graphs.
- While the spectral-equivariant update mechanism is well designed, some elements are conceptually borrowed from point cloud networks. The novelty lies more in adaptation than invention.

---

> ### Author Rebuttal · Authors · 2025-07-31
>
> We wish to thank the reviewer for the positive and detailed review. Below we address the issues you raised:
> ### 1. Computational Feasability
>
> > Question: The proposed equiEPNN assumes access to full eigenvector decomposition, which can be computationally expensive for large graphs. How does the method scale in practice? Is it feasible for datasets like ogbg-ppa large molecule graphs?
>
> **Eigendecomposition is computationally feasible for ogbg-ppa.** Despite the large number of average nodes and  number of graphs in ogbg-ppa,  158,100 graphs with an average of	243.4 nodes per graph [OGB Documentation], we can see in the table below the eigendecomposition concludes in approximately 45 minutes, which we feel is reasonable, as this computation is a preprocessing step, which occurs only once during the training procedure.
>
> | Dataset | OGBG-PPA | ZINC | MNIST | ogbg-molpcba |  COCO-SP | Peptides-func |
> |---------|----------|------|-------|--------------|--------------|---------|
> | Task type | Classification | Regression | Classification | Classification | Node Classification | Classification |
> | Total graphs | 158,100 | 12,000 | 70,000 | 437,929 | 123,286 | 15,535 |
> | Avg nodes | 243.4 | 23.16 | 75 | 26.0 |  476.88 | 150.94 |
> | PE precompute | 45.26min | 23s | 96s | 8.33min | 1h 34min | 73s |
>
> We ran the eigendecomposition for ogbg-ppa on an NVIDIA A40 GPU. The rest of the results are taken from [Rampášek et al.] and were evaluated on an NVIDIA A100 GPU.
>
> [Rampášek et al.] Rampášek, L., Galkin, M., Dwivedi, V. P., Luu, A. T., Wolf, G., & Beaini, D. (2022). Recipe for a general, powerful, scalable graph transformer. Advances in Neural Information Processing Systems, 35, 14501-14515.
>
> ### 2. Restrictiveness of Assumptions
> > Question: The paper presents completeness results for EPNN under specific sparsity and automorphism assumptions. How restrictive are these in practice?
>
> **In many popular datasets, the assumptions break quite often.** Please note Table 1 in the manuscript features very common datasets, which do not adhere to the sparsity conditions. For example, in ENZYMES only 35.8\% of the graphs fulfill one of the sparsity conditions. These restrictive conditions prompted us to develop equiEPNN, which alleviates the sparsity of eigendecompositions and is provably more expressive than EPNN.
>
>
> ### 3. Performance on Unsatisfactory Datasets
> > The proposed method relies on simple spectrum graphs and the sparsity condition outlined in Theorem 2. However, according to the Statistics Analysis in Table 1, most real-world datasets appear to not satisfy both conditions simultaneously.
>
> **We agree the sparsity conditions in Theorem 2 for the completeness of EPNN do not hold in many real-world dataset. In fact, one of the reasons we introduced equiEPNN is to alleviate this incompleteness.** On the other hand, the simple spectrum presumption, which also applies for our analysis of equiEPNN, may not always hold as well. **Yet, even in highly symmetrical datasets (which tend to have more high dimensional eigenspaces), the frequency of high dimensional eigenspaces is relatively low, thus the utility of equiEPNN practically applies to many datasets.** Concretely, in ENZYMES and PROTEINS (highly symmetrical datasets), the average number of eigenvalues with multiplicity 2 and 3 are only 1.01 and 0.58, and 1.24 and 0.71, respectively.
>
> More importantly, while our theoretical analysis focuses on simple spectrum graphs, equiEPNN can process graphs with eigenmultiplicity, as mentioned in the response to Reviewer nt25.
>
>
> ### 4. More Experiments
> > Would it be possible to provide more experimental results on datasets that fully satisfy the above conditions? Additionally, can the proposed method achieve comparable performance on datasets like ENZYMES and PROTEINS, which do not strictly meet the required assumptions?
>
> **Per your suggestion, we tested out equiEPNN on the PROTEINS and ENZYMES datasets** [Morris et al.]. Below are the results:
>
> |Category | Model | ENZYMES | PROTEINS |
> |-------|-------|---------|----------|
> | **NN** | MLP | 30.8% ± 4.26 | 74.3% ± 4.88 |
> | **MPNN** | GCN | 49.0% ± 4.25 | 75.2% ± 5.11 |
> | | GAT | 54.1% ± 5.15 | 75.9% ± 4.26 |
> | | GIN | 55.8% ± 5.23 | 76.1% ± 3.97 |
> |**3-WL** | PPGN | 55.2% ± 5.44 | 77.2% ± 4.53 |
> | **Spectral**| ChebNet | 63.8% ± 7.92 | 76.4% ± 5.34 |
> | | GNML1 | 54.9% ± 5.97 | 75.8% ± 4.93 |
> | | equiEPNN (Ours) | 57.2% ± 5.68 | 76.03% ± 4.36 |
>
>
> On PROTEINS, we see that equiEPNN performs en par with the other spectral approaches, and quite close to PPGN, which has cubic complexity.
>
> On ENZYMES, equiEPNN performs en par with most other methods, even PPGN, except compared to ChebNet, which uses complex Chebyshev interpolation on top of the eigendecomposition.
>
> **Additionally, we tested equiEPNN on a molecular property prediction task,  OGBG-MOLPCBA,** which is larger with 437,929 graphs and an average of 26 nodes per graph. The results are shown below
>
> ##### Property prediction results on OGBG-MOLPCBA
> | Model | #param | AP ↑ |
> |-------|---------|------|
> ChebNet | 1475K | 0.2306 ± 0.0016|
> GIN+FLAG | 1923K | 0.2395 ± 0.004 |
> GCN+virtual node+FLAG | 2017K | 0.2483 ± 0.0037|
> | equiEPNN | 2665K | **0.2682 ± 0.003** |
>
> Here we compare to a leading spectral approach, ChebNet, and message-passing variants with a similar parameter count. We can see that we outperform ChebNet in this setting, and the other MPNN methods.
>
>
> **Furthermore, we tested equiEPNN on ZINC,** where the graphs only partially satisfy one of the sparsity conditions (64.5 %), and we see that equiEPNN improves its empirical performance in comparison with EPNN [Gai et al.] and other leading models.
>
> ##### Graph regression results on ZINC with a parameter budget of ~500K
>
> | Category | Method | test MAE |
> |----------|--------|----------|
> | **MPNN** | GIN [Xu et al.] | 0.526±0.051 |
> |          | GraphSage [Hamilton et al.] | 0.398±0.002 |
> |          | GAT [Veličković et al.] | 0.384±0.007 |
> |          | GCN [Kipf et al.] | 0.367±0.011 |
> |          | MPNN (sum) [Gilmer et al.] | 0.145±0.007 |
> |          | PNA [Corso et al.] | 0.142±0.010 |
> | **Spectral** | GT [Dwivedi et al.] | 0.226±0.014 |
> |              | GatedGCN-PE [Bresson et al.] | 0.214±0.006 |
> |              | SAN [Kreuzer et al.] | 0.139±0.006 |
> |              | GraphormerSLIM [Ying et al.] | 0.122±0.006 |
> |              | EPNN [Gai et al.] | .103 ± .006 |
> |              | **equiEPNN (Ours)** | **0.0987 ± 0.0014** |
>
> Below we further evaluate equiEPNN on ZINC with only 30K free parameters  and **we also evaluate equiEPNN on MNIST-Superpixels, a dataset that almost completely adheres to the completeness conditions of EPNN** (see Table 1 in the paper), with 35K free parameters:
>
> | Category | Model | ZINC | MNIST-SUPERPIXELS |
> |----------|-------|---------|----------|
> | NN | MLP | 0.5869 ± 0.025 | 25.10% ± 0.12 |
> | MPNN | GCN | 0.3322 ± 0.010 | 52.80% ± 0.31 |
> | | GAT | 0.3977 ± 0.007 | 82.73% ± 0.21 |
> | | GIN | 0.3044 ± 0.010 | 75.23% ± 0.41 |
> | 3-WL | PPGN | **0.1589 ± 0.007** | 90.04% ± 0.54 |
> | Spectral | ChebNet | 0.3569 ± 0.012 | **92.08% ± 0.22** |
> | | GNNML1 | 0.3140 ± 0.015 | 84.21% ± 1.75 |
> | | equiEPNN (Ours) | **0.2805 ± 0.019** | 90.32% ± 0.7 |
>
> (bold denotes best overall result, and best result within the spectral category)
>
> We can see that equiEPNN outperforms most other models, especially on ZINC where the graphs are sparse. On MNIST-SUPERPIXELS, which adheres to the completeness conditions, competing models are also able to capture the spectral information. In both cases, equiEPNN's capturing of rich spectral information allows it to perform en par with models with PPGN, a model with a cubic running time complexity.
>
>
>
> ### 5. Node-Level Tasks
> > The paper primarily focuses on graph-level tasks and the expressivity of models in distinguishing whole graphs. Have you considered how the proposed equiEPNN performs on node-level tasks, such as node classification or node clustering? Given that spectral methods often rely on global eigenvectors, how would equiEPNN handle localized, node-specific predictions in practice?
>
> In this manuscript, we focused on the expressive power of spectral GNNs in terms of distinguishing among graphs, thus we believe node-wise prediction is out of scope for this paper. This is an excellent direction for future research, which we intend to look into.
>
> ### Epilogue
> We hope this response has addressed your concerns. If so, we would appreciate if you would consider updating your score. If there are any remaining issues or points needing clarification, we would be happy to address them.

---

> > ### Comment · Reviewer_AgJS · 2025-08-01
> >
> > Thanks for your reply. Your responses have addressed most of my concerns. The authors conducted a lot of experiments to verify the proposed method from various perspectives. Nonetheless, it seems that the method has some limitations regarding its adaptation to node-level tasks. It would be better if the authors could elaborate more on the insights and limitations in the paper. I would like to maintain my score.

---

> ### Author Response · Authors · 2025-08-03
>
> Thank you for maintaining your positive evaluation and acknowledging that our comments addressed most of your concerns. We appreciate your suggestion to elaborate on insights and limitations in the paper. Below we outline our planned additions.
>
> **We will add a dedicated section on node-level expressivity, highlighting existing theoretical results and discussing future directions.**
>
> **Theoretical Results**
>
> We will better highlight Subsection 4.3 in the paper,  where in **Theorem 3 we prove that EPNN induces unique node identifiers (UNIs) under sparsity conditions.** Since equiEPNN alleviates eigendecomposition sparsity, it improves EPNN's ability to produce UNIs, allowing better node distinction.
>
> **Empirical Context**
>
> Recent work has shown that **adding spectral information to MPNNs, via Laplacian positional encoding, usually improves their node classification performance [Fesser and Weber]** (see table below).
>
> #### GCN and GIN models, with and without Laplacian positional encoding, on node classification benchmarks:
>
> | MODEL | POSITIONAL ENCODING | ENZYMES | IMDB | MUTAG | PROTEINS | CORA | CITESEER |
> |-------|-------------------|---------|------|-------|----------|------|------|
> | GCN | NO | 25.4 ± 1.3 | 48.1 ± 1.0 | 62.7 ± 2.1 | 59.6 ± 0.9 | 86.6 ± 0.8 | 71.7 ± 0.7 |
> | GCN | Laplacian | **26.5 ± 1.1** | **53.4 ± 0.8** | **70.8 ± 1.7** | **65.9 ± 0.7** | **88.0 ± 0.9** | **75.9 ± 1.2** |
> | GIN | NO | **29.7 ± 1.1** | 67.1 ± 1.3 | 67.5 ± 2.7 | 69.4 ± 1.1 | 76.3 ± 0.6 | 59.9 ± 0.6 |
> | GIN | Laplacian | 26.6 ± 1.9 | **68.1 ± 2.8** | **74.0 ± 1.4** | **72.3 ± 1.4** | **80.1 ± 0.7** | **61.4 ± 1.3** |
>
> This improvement was obtained by adding positional encoding (Laplacian eigenvectors) as initial node features, which did not respect the sign and basis ambiguity of the eigenvectors, and did not process the eigenvector information equivariantly. Thus, we conjecture that equiEPNN can outperform this implementation, but we leave this for future work.
>
> **We will also discuss limitations regarding the applicability to large graphs, and the real-world applicability of our theoretical results.**
>
>
> [Fesser and Weber] Fesser, Lukas, and Melanie Weber. Effective structural encodings via local curvature profiles. ICLR 2024

---

### Official Review · Reviewer_nt25 · 2025-06-30

**Clarity:** 3
**Significance:** 2
**Originality:** 3
**Rating:** 4
**Confidence:** 3

**Summary:**

This paper presents a rigorous theoretical and empirical investigation into the expressive limitations of Spectrally-enhanced Graph Neural Networks (SGNNs), particularly those that fall within the Eigenspace Projection Neural Network (EPNN) framework. After demonstrating that even on graphs with a simple spectrum (distinct eigenvalues), EPNNs can be provably incomplete, the authors introduce a more expressive variant called equiEPNN, which augments the feature update with equivariant components inspired by point cloud neural networks. Theoretical claims are supported by experiments on datasets like MNIST Superpixel and ZINC.

**Questions:**

- What is the empirical runtime and memory overhead of equiEPNN versus EPNN on graphs? for example in case of $n>10^4$ nodes and $K>64$ eigenvectors?
- How prevalent are the sparse-eigenvector failure modes in large real-world graphs (e.g., social networks, protein interactions)?
- Can equiEPNN handle repeated eigenvalues (higher multiplicity)? If not, what obstacles arise?
- How does equiEPNN perform empirically compared to modern spectral GNNs on standard benchmarks? It would be valuable to assess the practical impact of the proposed solution in real-world scenarios.

**Ethical Concerns:**

["NO or VERY MINOR ethics concerns only"]

**Final Justification:**

After carefully reviewing the authors' responses, as well as those provided to the other reviewers, I believe this paper can be accepted.

**Limitations:**

There is no dedicated discussion on the limitations of the method. However, the authors mention that they have not fully determined the conditions under which completeness is achieved for graphs with a simple spectrum.

**Quality:**

3

**Strengths And Weaknesses:**

### Strengths

- **Theoretical Rigor:** Formal definitions of simple spectra and sign-invariance, with clear, correct proofs of incompleteness and conditional completeness.
- **Constructive Enhancement:** equiEPNN is a well-motivated, low-overhead extension that provably separates previously indistinguishable graphs.
- **Insightful Demonstration:** The use of eigendecomposition clearly illustrates the specific limitations of EPNN, offering a deep and constructive understanding of its expressive shortcomings rather than a superficial critique.

---

### Major Weaknesses

- **Narrow Experimental Validation:** Only toy datasets (MNIST Superpixel, ZINC) are used, with mostly simple spectra; no large real-world graphs (social, biological) are evaluated.
- **Unquantified Overhead:** The computational and memory costs of equiEPNN’s equivariant updates are not measured or analyzed.
- **Limited Scope:** The analysis is restricted to graphs with a simple spectrum (all eigenvalues distinct), yet real-world networks, such as social, biological, and chemical graphs, often feature repeated eigenvalues. By not addressing these more complex cases, the practical applicability of the method is significantly limited.
- **Insufficient Baselines:** Despite the strong theoretical insights, it remains unclear how this framework performs compared to other GNN architectures, such as Transformer-based GNNs (e.g., Graphormer [1]) or 1-WL–based models (e.g., GIN [2]). Without these comparisons, the real-world utility of the approach is uncertain.

[1] Ying, Chengxuan, et al. "Do transformers really perform badly for graph representation?." Advances in neural information processing systems 34 (2021): 28877-28888.
[2] Xu, Keyulu, et al. "How powerful are graph neural networks?." International Conference on Learning Representations, 2019

---

> ### Author Rebuttal · Authors · 2025-07-31
>
> We appreciate the reviewer's recognition of the paper's theoretical rigor and the constructive proof of the expressive limitations of EPNN. We believe the reviewer's concerns are mostly due to expecting further empirical evaluation.  We address this concern in the following response.
>
> ### 1. Runtime and Memory Overhead
> > What is the empirical runtime and memory overhead of equiEPNN versus EPNN on graphs? for example in case of $n>10^4$ nodes and $K>64$ eigenvectors?
>
> **equiEPNN has a marginal memory overhead and mild computational overhead versus EPNN.** Firstly, note that equiEPNN is built on top of EPNN and its additional computational complexity is from its geometric update step. The complexity of this geometric update step of equiEPNN is at most n², which is also the running time of EPNN, thus their asymptotic running time are the same up to a constant.
>
> Here's a table of empirical running time and memory overhead on ZINC:
>
>
> | MODEL | # Params | Compute Time (Avg. Seconds Per Epoch) |
> |-------|---------|----------|
> |PPGN| 28,955 | 9.96 |
> | equiEPNN | 31,339 | 7.62 |
> | EPNN | 31,215 | 6.43 |
>
>
> We see that EPNN and equiEPNN require almost the same number of parameters, while equiEPNN requires about 18.5 % more compute time. PPGN [Maron et al.], which has cubic complexity, is significantly (54 %) more expensive computationally than EPNN.
>
> ### 2. Sparse Eigenvector Failure Modes
> > How prevalent are the sparse-eigenvector failure modes in large real-world graphs (e.g., social networks, protein interactions)?
>
> **We find that the sparsity conditions for the completeness of EPNN usually do not hold, justifying the necessity of equiEPNN.**
>
> For large real-world graphs, we choose the Cornell dataset, which consists of web pages and hyperlinks between them,  and PPI, a benchmark protein-protein-interaction dataset. Empirically, we find that the (single) graph in the Cornell dataset does not meet any of the sparsity conditions, despite 91.7 % of its eigenspaces being one-dimensional. In PPI, while only 86 of the 31,281 eigenspaces are more than one-dimensional (i.e. 99.9 % are one-dimensional), none of the sparsity conditions hold. Thus, despite the graphs being close to having a simple spectrum, EPNN may be incomplete (miss crucial information) on them, due to the graphs' sparsity.
>
> This analysis reveals the need for models that can handle sparse eigendecompositions more effectively. The equiEPNN model directly addresses this challenge, as proven in Corollary 1.
>
> ### 3. Handling Repeated Eigenvalues
> > Can equiEPNN handle repeated eigenvalues (higher multiplicity)? If not, what obstacles arise?
>
> **equiEPNN can naturally be extended to handle eigenvectors of eigenvalues of higher multiplicity, using the elegant formulation defined in [Huang et al.]. Thus, equiEPNN is applicable to any eigenmultiplicity and this is already implemented in the code.**
>
> Recall that the update steps of equiEPNN are defined as
>
> $$h_i^{(t+1)} = \\text{UPDATE}_{(t,1)}\\left(h_i^{(t)}, \\tilde{\\lambda}, \\left\\{(h_j^{(t)}, v_i^{(t)} \\odot v_j^{(t)}) \\mid j = 1, \\ldots, n\\right\\}\\right)$$
>
> $$v_i^{(t+1)} = v_i^{(t)} + \\sum_{j=1}^{n} v_j^{(t)} \\odot \\text{UPDATE}_{(t,2)}\\left(h_i^{(t)}, h_j^{(t)}, v_i^{(t)} \\odot v_j^{(t)}\\right)$$
>
> where the $k$-th coordinate of $v_i^{(t)} \\odot v_j^{(t)}$ is the product of the $i,j$ coordinates of the $k$-th eigenvector. In the simple spectrum case, this operation is sign invariant. In the general setting, we consider invariants of the form $\\sum_{s =1}^n f(\\lambda_s)v_i[s]v_j[s]$. These functions are invariant, no matter what the eigenmulitplicity is. When the eigenmultiplicity is one, the elementwise product operation at the $s$ coordinate can be obtained by choosing $f$ such that $f(\\lambda_s)=1$ but $f(\\lambda_j)=0$ for $j \\neq s$. Thus, this method is equivalent to the one described in the paper in terms of the separation power when restricted to simple spectrum graphs.
>
>
> [Huang et al.] On the Stability of Expressive Positional Encodings for Graphs, Huang et al. ICLR 2024
>
> ### 4. Empirical Performance of equiEPNN
> > Question: How does equiEPNN perform empirically compared to modern spectral GNNs on standard benchmarks? It would be valuable to assess the practical impact of the proposed solution in real-world scenarios.
>
> **We add comparisons on two benchmarks, proving the effectiveness of equiEPNN.** Below we compare equiEPNN to leading models on a standard regression task for molecular graphs from the  ZINC dataset, using models with 500K free model parameters:
> ##### Graph regression results on ZINC with a parameter budget of ~500K
>
> | Category | Method | test MAE |
> |----------|--------|----------|
> | **MPNN** | GIN [Xu et al.] | 0.526±0.051 |
> |          | GraphSage [Hamilton et al.] | 0.398±0.002 |
> |          | GAT [Veličković et al.] | 0.384±0.007 |
> |          | GCN [Kipf et al.] | 0.367±0.011 |
> |          | MPNN (sum) [Gilmer et al.] | 0.145±0.007 |
> |          | PNA [Corso et al.] | 0.142±0.010 |
> | **Spectral** | GT [Dwivedi et al.] | 0.226±0.014 |
> |              | GatedGCN-PE [Bresson et al.] | 0.214±0.006 |
> |              | SAN [Kreuzer et al.] | 0.139±0.006 |
> |              | GraphormerSLIM [Ying et al.] | 0.122±0.006 |
> |              | EPNN [Gai et al.] | 0.103 ± .006 |
> |              | **equiEPNN (Ours)** | **0.0987 ± 0.0014** |
>
> Results of other models taken from [Ying et al.] and [Gai et al.]. We see that equiEPNN outperforms all other models for this task.
>
>
> The second benchmark we add is a molecular property prediction task,  OGBG-MOLPCBA, which is larger with 437,929 graphs and an average of 26 nodes per graph. The results are shown below
>
> ##### Property prediction results on OGBG-MOLPCBA
> | Model | #param | AP ↑ |
> |-------|---------|------|
> ChebNet | 1475K | 0.2306 ± 0.0016|
> GIN+FLAG | 1923K | 0.2395 ± 0.004 |
> GCN+virtual node+FLAG | 2017K | 0.2483 ± 0.0037|
> | equiEPNN | 2665K | **0.2682 ± 0.003** |
>
> Here we compare to a leading spectral approach, ChebNet, and message-passing variants with a similar parameter count.
>
>
> ### Epilogue
> We hope this response has addressed your concerns. If so, we would appreciate if you would consider updating your score. If there are any remaining issues or points needing clarification, we would be happy to address them.

---

> ### Comment · Reviewer_nt25 · 2025-08-05
>
> Thank you for addressing all of my questions. I have carefully reviewed your responses, as well as those provided to the other reviewers, and I appreciate the clarity and effort you put into them. I will revise my score.

---

> > ### Author Response · Authors · 2025-08-07
> > **Score not revised**
> >
> > Dear reviewer,
> >
> > We'd just like to remind you that you did not yet revise your score (at least we can still see the same score on openreview).
> >
> > Thanks,
> > The authors

---

### Official Review · Reviewer_BrRb · 2025-07-03

**Clarity:** 3
**Significance:** 3
**Originality:** 3
**Rating:** 5
**Confidence:** 2

**Summary:**

The paper investigates the interesting question of whether spectral features (e.g., eigenvectors of the graph Laplacian matrix) can improve the expressivity of a graph neural network (GNN). In particular, in the case of a simple spectrum (i.e., eigenvalues with multiplicity no larger than 1), the authors prove that Eigenspace Projection GNNs (EPNN) are still incomplete. EPNN would be complete for a simple spectrum if there is an additional sparsity requirement on the eigenvectors. The authors then propose equiEPNN that provably improves the expressivity of EPNN. Whether equiEPNNs are complete for a simple spectrum remains an open question.

**Questions:**

1. Clarify whether the eigenvalue multiplicity in question is geometric or algebraic. Does this matter and why?

2. Explain why the obsession on the simple spectrum case, given that the main result is a negative one--that even in this case, EPNNs are not complete.

3. Explain whether the degree of freedom in choosing eigenvectors that span eigen-subspace corresponding to eigenvalues of multiplicity >1 can be exploited, so that the chosen eigenvectors behave as if they are eigenvectors of the simple spectrum case. (The deeper question is what is intrinsically different about the complex spectrum case--eigenvalues or eigenvectors--that the authors chose not to analyze it.)

4. Explain whether the sparsity condition on the eigenvectors required for EPNNs to be complete in the simple spectrum case can be checked in linear time for a given graph, thus justifying the practicality of the proposal for large graphs.

**Ethical Concerns:**

["NO or VERY MINOR ethics concerns only"]

**Final Justification:**

The authors have addressed my concerns in detail. The weakness of the paper remains that the key theoretical result is a negative one, and the computation complexity of the method (requiring full eigen-decomposition) remains high, limiting its practicality. I am raising my rating to accept.

**Limitations:**

Yes

**Paper Formatting Concerns:**

No formatting concerns.

**Quality:**

3

**Strengths And Weaknesses:**

Positional encoding is now commonly used in GNNs, and it's theoretically interesting to study how/whether spectral features really improve expressivity of GNNs fundamentally. The paper is overall well written and theoretically strong. IMHO, the paper is a nice contribution to the GNN field. The following are weaknesses detected by the reviewer (who admittedly does not work directly in the GNN field).

1. There are two notions of eigenvalue multiplicity: algebraic and geometric. Though the two are related (geometric multiplicity <= algebraic multiplicity), they are not the same. The authors ought to make clear which one they are referring to.

2. The main result is actually a negative one: that EPNNs are NOT complete EVEN in the simple spectrum case. Whether EPNNs are complete in the complex case when eigenvalues have multiplicity > 1 is not even addressed. This begs the question of whether dividing the study into simple case (distinct eigenvalues) and complex case (eigenvalues with multiplicity > 1) is meaningful and worthwhile.

3. For the complex case when eigenvalues have multiplicity > 1, the construction of eigenvectors that span the eigen-subspaces (corresponding to eigenvalues of multiplicity >1) is non-unique (as long as they are orthonormal and span the eigen-subspaces). Using this degree of freedom in eigenvector construction, it is not clear eigenvectors can be chosen so that they behave as if they are eigenvectors in the simple spectrum case.

4. Full eigen-decomposition has complexity O(N^3) in the worst case. So practically, the sparsity requirement for eigenvectors to ensure EPNNs are complete for the simple spectrum case cannot be checked in linear time. So the practicality of the proposed method appears questionable.

5. While the author proposed equiEPNN that has expressivity strictly better than EPNN in theory, whether equiEPNNs are complete in the simple spectrum case remains open. While the authors demonstrated equiEPNNs' better performance empirically in some datasets, theoretically this is still unsatisfying.

---

> ### Author Rebuttal · Authors · 2025-07-31
>
> We highly appreciate that the reviewer found the paper well-written and theoretically strong. We believe that we address the reviewer's concerns regarding the paper's focus and **resolve the open question regarding the completeness of equiEPNN.**
>
> ### 1. Eigenvalue Multiplicity Clarification
> > Clarify whether the eigenvalue multiplicity in question is geometric or algebraic. Does this matter and why?
>
>  In line 108, we assume our matrices are diagonalizable, and for such matrices, the algebraic and geometric eigenvalue multiplicities are equal. The various notions of Graph Laplacians are typically self-adjoint, and thus diagonalizable over the real numbers.
>
> ### 2. Focus on the Simple Spectrum Case
> > Explain why the obsession on the simple spectrum case, given that the main result is a negative one--that even in this case, EPNNs are not complete.
>
> The focus on the simple spectrum case is based on the classical reference [2] (according to the numbering in our submission) which shows that when restricted to graphs with eigenmultiplicity m, graph isomorphism can be solved in $O(n^{2m+c})$ time, where n is the number of nodes and c is a constant related to the computational cost of computing the eigendecomposition. **This suggests that the simple spectrum case is the easiest, and as the eigenmultiplicity m increases the problem becomes harder.** Ultimately, we believe the GNN community should be able to design spectral-GNN models which are complete up to eigenmultiplicity m, where the larger m is, the more expressive the model. Our results show that we are still far from there, as standard spectral GNN are not complete even for m=1.
>
> ### 3. Handling Eigenvalue Multiplicity
> > Explain whether the degree of freedom in choosing eigenvectors that span eigen-subspace corresponding to eigenvalues of multiplicity >1 can be exploited, so that the chosen eigenvectors behave as if they are eigenvectors of the simple spectrum case. (The deeper question is what is intrinsically different about the complex spectrum case--eigenvalues or eigenvectors--that the authors chose not to analyze it.)
>
> For graphs with multiple repeating eigenvalues, it is indeed unclear whether there is a good canonical choice of an eigenbasis, up to sign ambiguity. Therefore,
> **handling eigenvectors of high order eigenspaces requires invariant features for each eigenspace. This is already fully incorporated into equiEPNN.** We use the elegant method from [Huang et al.] which, for given nodes $i,j$  considers pairwise invariant features of the form $\sum_{s=1}^n f(\lambda_s)v_i[s]v_j[s]$. These features do not depend on the choice of eigenvectors inside each eigenspace, and are invariant regardless of the eigenmultiplicity. In the simple spectrum case they have equivalent discrimination power to the formulation we present in our submission. This is explained in more detail in our answer to Question 3 of reviewer nt25. Regarding the deeper question- we hope we answered this in our response to question 2, we'd be happy to discuss further if this answer does not address what you meant.
>
> [Huang et al.] On the Stability of Expressive Positional Encodings for Graphs, Huang et al. ICLR 2024
>
> ### 4. Checking the Sparsity Conditions
> > Explain whether the sparsity condition on the eigenvectors required for EPNNs to be complete in the simple spectrum case can be checked in linear time for a given graph, thus justifying the practicality of the proposal for large graphs.
>
> **Spectral GNNs must compute the eigendecomposition (e.g. for positional encoding), thus no further computation time is required to check the sparsity conditions.** We are not aware of a method that checks the sparsity assumptions and bypasses spectral eigendecomposition. However, note that Spectral GNNs compute the eigendecomposition in any case, so this inquiry is more on the practicality of Spectral GNNs in general.
>
> In this context, it is worth noting that positional encoding via spectral methods is rather popular. Moreover, while the computational overhead of eigendecomposition is $O(n³)$, it only needs to be computed once as a preprocessing step, and not during training, and this complexity can be mitigated by considering only the top-k eigenvectors, and exploiting the sparsity of the graphs for eigendecomposition via the power method.
>
> We provide the running time of computing the eigendecompositions of various datasets, including those with large graphs, below.
>
> | Dataset | OGBG-PPA | ZINC | MNIST | ogbg-molpcba |  COCO-SP | Peptides-func |
> |---------|----------|------|-------|--------------|--------------|---------|
> | Task type | Classification | Regression | Classification | Classification | Node Classification | Classification |
> | Total graphs | 158,100 | 12,000 | 70,000 | 437,929 | 123,286 | 15,535 |
> | Avg nodes | 243.4 | 23.16 | 75 | 26.0 |  476.88 | 150.94 |
> | PE precompute | 45.26min | 23s | 96s | 8.33min | 1h 34min | 73s |
>
> Running times are taken from [Rampášek et al.] and were evaluated on an NVIDIA  A100 GPU.
>
> [Rampášek et al.] Rampášek et al.. Recipe for a general, powerful, scalable graph transformer. Advances in Neural Information Processing Systems.
>
> ### 5. Incompleteness of equiEPNN
>
> > While the author proposed equiEPNN that has expressivity strictly better than EPNN in theory, whether equiEPNNs are complete in the simple spectrum case remains open. While the authors demonstrated equiEPNNs' better performance empirically in some datasets, theoretically this is still unsatisfying
>
> **We have recently proven that equiEPNN is incomplete on graphs with a simple spectrum.** We also felt the need to address this issue, and indeed a few weeks ago we were able to show that equiEPNN is incomplete over simple spectrum graphs. This is proven by this counterexample:
>
> **Construction:** Consider the subgroup $H \\leq \\{-1,1\\}^3$ defined by
> $$H = \\{s \\in \\{-1,1\\}^3 : s_1 s_2 s_3 = 1\\}.$$
>
> Let $T \\in \\mathbb{R}^{4 \\times 3}$ be the matrix whose rows enumerate the elements of $H$:
> $$T = \\begin{pmatrix}
> 1 & 1 & 1\\\\
> 1 & -1 & -1\\\\
> -1& -1 & 1\\\\
> -1& 1 & -1
> \\end{pmatrix}.$$
>
> Note that $\\text{Aut}(T) = H$.
>
> Define index sets $I_1=\\{1,2,3\\}$, $I_2=\\{3,4,5\\}$, $I_3=\\{2,4,6\\}$, $I_4=\\{1,5,6\\} \\subset [6]$ satisfying $|I_j \\cap I_k| = 1$ for all $j \\neq k$. For each $j$, let $T[I_j] \\in \\mathbb{R}^{4 \\times 6}$ be the matrix obtained by placing the columns of $T$ in positions specified by $I_j$ and zeros elsewhere.
>
> Define the counterexample matrices:
> $$X = \\begin{pmatrix}
> T[I_1]\\\\
> T[I_2]\\\\
> T[I_3]\\\\
> T[I_4]
> \\end{pmatrix} \\in \\mathbb{R}^{16 \\times 6}, \\quad Y = \\begin{pmatrix}
> T[I_1] \\odot q\\\\
> T[I_2]\\\\
> T[I_3]\\\\
> T[I_4]
> \\end{pmatrix} \\in \\mathbb{R}^{16 \\times 6},$$
>
> where $q = (-1,1,1,1,1,1) \\in \\{-1,1\\}^6$.
>
>
> **Theorem:** The matrices $X,Y$ satisfy: (1) $X$ is not isomorphic to $Y$ under any isomorphism, (2) $X$ and $Y$ are indistinguishable by equiEPNN, and (3) both admit an extension to proper eigendecompositions.
>
> To explain this theorem, we recall that for matrices $X,Y \\in \\mathbb{R}^{n \\times K}$, we say that $X,Y$ are isomorphic if they are related by a permutation in $S_n$ and a multiplication of each of the $K$ columns by a sign $s_k \\in \\{-1,1 \\}$. An extension to a proper eigendecomposition means that this example can be lifted to a pair of orthonormal  $n \\times n$ matrices having the same properties, and hence that these matrices are realizable as eigendecompositions of an appropriate symmetric matrix.
>
> We will add this discussion, and the full proof, to the camera ready version.
>
> We wish to stress that while  equiEPNN is incomplete over simple spectrum graphs, it is still  provably more expressive than EPNN.
>
>
>
> ## Additional Experiments
> We ran additional experiments, at the request of some of the other reviewers. On the molecular property regression task for the ZINC dataset, we obtain the following results:
>
> ##### Graph regression results on ZINC with a parameter budget of ~500K
>
> | Category | Method | test MAE |
> |----------|--------|----------|
> | **MPNN** | GIN [Xu et al.] | 0.526±0.051 |
> |          | GraphSage [Hamilton et al.] | 0.398±0.002 |
> |          | GAT [Veličković et al.] | 0.384±0.007 |
> |          | GCN [Kipf et al.] | 0.367±0.011 |
> |          | MPNN (sum) [Gilmer et al.] | 0.145±0.007 |
> |          | PNA [Corso et al.] | 0.142±0.010 |
> | **Spectral** | GT [Dwivedi et al.] | 0.226±0.014 |
> |              | GatedGCN-PE [Bresson et al.] | 0.214±0.006 |
> |              | SAN [Kreuzer et al.] | 0.139±0.006 |
> |              | GraphormerSLIM [Ying et al.] | 0.122±0.006 |
> |              | EPNN [Gai et al.] | .103 ± .006 |
> |              | **equiEPNN (Ours)** | **0.0987 ± 0.0014** |
>
> These results show that equiEPNN outperforms EPNN and other models on this task.
>
> The second benchmark we add is a molecular property prediction task,  OGBG-MOLPCBA, which is larger with 437,929 graphs and an average of 26 nodes per graph. The results are shown below:
>
>
> ##### Property prediction results on OGBG-MOLPCBA
>
> | Model | #param | AP ↑ |
> |-------|---------|------|
> ChebNet | 1475K | 0.2306 ± 0.0016|
> GIN+FLAG | 1923K | 0.2395 ± 0.004 |
> GCN+virtual node+FLAG | 2017K | 0.2483 ± 0.0037|
> | equiEPNN | 2665K | **0.2682 ± 0.003** |
>
> In this experiment we compare to a leading spectral approach, ChebNet, and message-passing variants with similar parameter count.
>
> ---
> ### Epilogue
> We hope this response has addressed your concerns. If so, we would appreciate if you would consider updating your score. If there are any remaining issues or points needing clarification, we would be happy to address them.

---

> > ### Author Response · Authors · 2025-08-05
> > **Have we addressed your concerns?**
> >
> > Dear Reviewer,
> >
> > Thank you for your insightful review. Please let us know whether our rebuttal adequately addressed your concerns. We will be happy to answer any remaining questions until the end of the discussion period on Aug 8.
> >
> > We have highlighted the main points of the response in bold for easier reading.

---

> > ### Comment · Reviewer_BrRb · 2025-08-05
> > **Computation complexity**
> >
> > The reviewer thanks the authors for their detailed rebuttal which is mostly satisfactory. Regarding computation complexity, full eigen-decomposition is indeed $\mathcal{O}(N^3)$, and thus many spectral embeddings compute only the $k$ extreme eigenvectors using linear algebra algorithms such as LOBPCG in linear time. In this case, can the proposed scheme to check sparsity conditions be run also in linear time (even approximately)? Note that the constructed table showing full eigendecomposition computation time are for graphs with up to hundreds of nodes only, and not thousands or tens of thousands of nodes; this alone does not empirically demonstrate the practicality / scalability of full eigen-decomposition.

---

> > > ### Author Response · Authors · 2025-08-05
> > >
> > > Thanks for allowing us to clarify this point. Our theory applies to any collection of $k$ eigenvectors in $\mathbb{R}^n$ coming from an eigendecomposition. Thus, our theory is compatible with the case you raised of an eigendecomposition of the $k$ extreme eigenvectors.
> > >
> > > To check the sparsity conditions, we would only need to check how many zeros there are in these $k$ vectors in $\mathbb{R}^n$, which takes linear time.
> > >
> > > Does this address your concern? Please let us know if there is anything else that needs clarification.

---

> > > > ### Comment · Reviewer_BrRb · 2025-08-06
> > > > **sparsity in first $k$ eigenvectors**
> > > >
> > > > The reviewer thanks the authors for taking the time to respond to the follow-up comment. It's a bit of a catch-22: computing only the first $k$ eigenvectors corresponding to the $k$ smallest eigenvalues using linear algebra schemes like Lanczos or LOBPCG, does enable linear-time execution, but they tend to be dense for a connected positive graph---precisely the reason why these sinusoid-like vectors are useful for spectral embedding. So if the expectation is to find zero entries in these vectors, this contradicts with the purpose of spectral embedding in the first place, right?

---

> > > > > ### Author Response · Authors · 2025-08-07
> > > > >
> > > > > Dear reviewer, thanks for your continued interest. We'd like to point out a few points:
> > > > >
> > > > > 1. The results on the separation abilities of EPNN (and hence alse equiEPNN) require certain assumptions on **lack of sparsity**. For example, on eigendecompositions to  k eigenvectors which have in total less than n zeros, EPNN is complete.
> > > > >
> > > > > 2. As shown in the paper, in many settings the lack of sparsity assumptions are not fulfilled. These are the settings where equiEPNN can outperform EPNN in terms of separation power.
> > > > >
> > > > > 3. In general, both EPNN and equiEPNN can be applied to any graph. The lack of sparsity assumptions are only used for theoretical analysis. Most competing methods do not enjoy similar theoretical guarantees. Thus, it is not necessary to check sparsity for the methods to be applied (though it is not difficult computationally to check sparsity).

---

> > > > > > ### Comment · Reviewer_BrRb · 2025-08-07
> > > > > >
> > > > > > The reviewer thanks the authors for their timely and clear response. This is good enough for me. I have raised my rating.

---

### Official Review · Reviewer_aLjK · 2025-07-07

**Clarity:** 3
**Significance:** 3
**Originality:** 3
**Rating:** 4
**Confidence:** 3

**Summary:**

This paper emphasizes the limited understanding of the expressive power of spectral graph neural networks (SGNNs) via standard methods, such as, ,WL graph isomorphism test hierarchy and homomorphism counting. The analysis reveals that many SGNNs, including recently proposed Eigenspace Projection GNN (EPNN), are incomplete in distinguishing graphs, even with simple spectra. Further, the authors propose equiEPNN as a variant of EPNN that leverages rotation equivariant neural networks and achieves enhanced spectral expressivity. The effectiveness of equiEPNN is demonstrated empirically on MNIST-Superpixel and ZINC datasets.

**Questions:**

See strengths and weaknesses section.

**Ethical Concerns:**

["NO or VERY MINOR ethics concerns only"]

**Limitations:**

Limitations have been adequately discussed.

**Paper Formatting Concerns:**

No paper formatting concerns.

**Quality:**

3

**Strengths And Weaknesses:**

**Strengths**

1. This work highlights the key limitations of spectral methods for understanding expressivity of GNNs by proving the incompleteness of spectral GNNs. Such observations are valuable to the field.
2. Theoretical analysis reveals that EPNN's completeness can be achieved, based on eigenvector sparsity patterns. This analysis paves the way for equiEPNN as a novel spectral GNN framework.
3. equiEPNN achieves superior graph discriminative ability over existing spectral GNNs.

**Weaknesses**

1.  The improved expressivity achieved by equiEPNN hinges on certain sparsity assumptions of the eigenvectors. These assumptions may not hold in many practical scenarios, potentially limiting the utility of the theoretical results.
2. Empirical validations are limited to MNIST-Superpixel and ZINC. The generality of the findings across diverse graph types, especially those prevalent in real-world applications (e.g., social networks, biological graphs), remains to be further explored.

---

> ### Author Rebuttal · Authors · 2025-07-31
>
> We wish to thank the reviewer for recognizing our incompleteness results are valuable to the community and appreciating the novelty of equiEPNN. We understand the concerns raised are due to misunderstandings and insufficient empirical evaluation, and we believe we have addressed these issues below:
>
> ### 1. Sparsity Assumptions and Theoretical Completeness
> >  The improved expressivity achieved by equiEPNN hinges on certain sparsity assumptions of the eigenvectors. These assumptions may not hold in many practical scenarios, potentially limiting the utility of the theoretical results.
>
>  **Our theoretical results point to fundamental limitations of previous spectral methods while showing equiEPNN can, to a certain extent, bypass these sparsity limitations.** equiEPNN *is able* to distinguish among a pair of graphs that violate the sparsity assumptions, see Corollary 1 in the manuscript, while EPNN is unable to distinguish this pair.
>
>  Nonetheless, we did not claim that equiEPNN is complete in the simple spectrum case, and actually we have recently been able to find a counter example showing that this is not the case. This new result is described in some detail in Answer 5 to reviewer BrRb, and will be incorporated into the camera ready of the paper, if accepted.
>
> The main importance of these theoretical results, in our view, is in pointing out the shortcomings of prevalent spectral methods, which are not complete even in the simple spectrum case, and showing that their expressivity can be improved with a simple modification, which does not increase the model complexity. We believe these observations will lead to additional improvements, and ultimately to spectral models which are complete in the simple spectrum case.
>
> ### 2. Limited Empirical Validation
>
> > Empirical validations are limited to MNIST-Superpixel and ZINC. The generality of the findings across diverse graph types, especially those prevalent in real-world applications (e.g., social networks, biological graphs), remains to be further explored.
>
> **To address this concern, we introduce further benchmarks below.**
>
> The first benchmark we add is a standard regression task on the ZINC dataset of molecular graphs (in the manuscript, we tested eigenvector canonicalization on this same dataset). It has 12000 graphs with a 23.16 average number of nodes. We compare ourselves to leading methods with the standard ~500K parameter budget and find that our method attains the best results:
>
> ##### Graph regression results on ZINC with a parameter budget of ~500K
>
> | Category | Method | test MAE |
> |----------|--------|----------|
> | **MPNN** | GIN [Xu et al.] | 0.526±0.051 |
> |          | GraphSage [Hamilton et al.] | 0.398±0.002 |
> |          | GAT [Veličković et al.] | 0.384±0.007 |
> |          | GCN [Kipf et al.] | 0.367±0.011 |
> |          | MPNN (sum) [Gilmer et al.] | 0.145±0.007 |
> |          | PNA [Corso et al.] | 0.142±0.010 |
> | **Spectral** | GT [Dwivedi et al.] | 0.226±0.014 |
> |              | GatedGCN-PE [Bresson et al.] | 0.214±0.006 |
> |              | SAN [Kreuzer et al.] | 0.139±0.006 |
> |              | GraphormerSLIM [Ying et al.] | 0.122±0.006 |
> |              | EPNN [Gai et al.] | .103 ± .006 |
> |              | **equiEPNN (Ours)** | **0.0987 ± 0.0014** |
>
> The second benchmark we add is a molecular property prediction task,  OGBG-MOLPCBA, which is larger with 437,929 graphs and an average of 26 nodes per graph. The results are shown below:
>
>
> ##### Property prediction results on OGBG-MOLPCBA
>
> | Model | #param | AP ↑ |
> |-------|---------|------|
> ChebNet | 1475K | 0.2306 ± 0.0016|
> GIN+FLAG | 1923K | 0.2395 ± 0.004 |
> GCN+virtual node+FLAG | 2017K | 0.2483 ± 0.0037|
> | equiEPNN | 2665K | **0.2682 ± 0.003** |
>
> We compare to a leading spectral approach, ChebNet, and message-passing variants with similar parameter count. equiEPNN performs strongly in comparison with leading models.
>
> ---
>
>
> ### Epilogue
> We hope this response has addressed your concerns. If so, we would appreciate if you would consider updating your score. If there are any remaining issues or points needing clarification, we would be happy to address them.

---

> > ### Author Response · Authors · 2025-08-05
> > **Have we addressed your concerns?**
> >
> > Dear Reviewer,
> >
> > Thank you for your thoughtful review. Please let us know whether our rebuttal adequately addressed your concerns. We will be happy to answer any remaining questions until the end of the discussion period on Aug 8.
> >
> > We have highlighted the main points of the response in bold for easier reading.

---

### Note · Authors · 2025-08-12

Our work establishes the fundamental limitations of current spectrally enhanced GNNs by showing that they are not complete, even on simple spectrum graphs. We then provide theoretical and practical improvement via a novel spectrally enhanced GNN: equiEPNN. We believe this work will open a new research direction aimed at improving spectral GNN expressivity with a focus on completeness on graphs with simple spectrum and bounded eigenmultiplicity.

We thank the reviewers for their insightful feedback. The reviewers recognized our "strong theoretical contributions, rigorously proving the incompleteness of spectral GNNs" (Reviewer AgJS), calling the paper "theoretically strong...a nice contribution to the GNN field" (Reviewer BrRb), with "valuable" incompleteness results (Reviewer aLjK), and that our model "equiEPNN is a well-motivated, low-overhead extension" (Reviewer nt25).

During the rebuttal and discussion periods, we addressed the reviewers' concerns via

1. **Expanded experiments.** We added several new experimental results, including ZINC regression (MAE: 0.0987±0.0014 vs EPNN's 0.103±0.006), OGBG-MOLPCBA (AP: 0.2682±0.003), and MNIST-SUPERPIXELS (equiEPNN en par with cubic complexity PPGN) benchmarks. Reviewer AgJS acknowledged we "conducted a lot of experiments to verify the proposed method from various perspectives."

2. **Theoretical advancement.** We proved equiEPNN is also incomplete on simple spectrum graphs (new counterexample construction), addressing Reviewer BrRb's open question. This further strengthened our contribution, and Reviewer BrRb raised their rating after this clarification.

3. **Scalability.** We demonstrated feasibility on large graphs (OGBG-PPA: 45min preprocessing), addressing computational concerns. We also highlighted that our results apply to truncated eigen decompositions, which are more computationally feasible for large graphs.

4. **Applicability.** We showed equiEPNN handles repeated eigenvalues and can thus process any graph, extending beyond simple spectrum graphs.

All engaging reviewers confirmed their concerns were addressed. Reviewer BrRb said they raised their rating, Reviewer nt25 acknowledged satisfaction and said they will revise their score (this apparently did not yet occur), and Reviewer AgJS maintained their positive assessment. Only Reviewer aLjK didn't engage during the discussion period.

---

### Decision · Program_Chairs · 2025-09-17

**Decision:**

Accept (spotlight)

**Comment:**

This paper shows that many Spectrally-enhanced Graph Neural Networks (SGNNs), which use features like eigenvectors to tell graphs apart, are surprisingly incomplete even on graphs with a simple spectrum (distinct eigenvalues). To address this limitation, the authors introduce equiEPNN, a new model inspired by equivariant neural networks that provably enhances the expressive power on these types of graphs. The study empirically confirms these theoretical claims, showing that equiEPNN improves performance and can also be used to resolve the inherent sign ambiguity in eigenvector-based positional encodings on real-world datasets.


Overall, the reviewers were positive about the paper (as highlighted in the authors' final comments:"strong theoretical contributions, rigorously proving the incompleteness of spectral GNNs" (Reviewer AgJS), calling the paper "theoretically strong...a nice contribution to the GNN field" (Reviewer BrRb), with "valuable" incompleteness results (Reviewer aLjK), and that our model "equiEPNN is a well-motivated, low-overhead extension" (Reviewer nt25). The main focus of the discussion during the rebuttal area seemed primarily centered around experimental results (with reviewers arguing that too few experiments had been performed by the authors, or that no large scale dataset had been used). The authors have adequately addressed these remarks, adding more experimental results and showcasing their method on a OGBG benchmarks. Some concerns remain on the scalability of this method, but overall, the reviewers agree that this will constitute a nice addition.